# Biological Mechanisms to Reduce Radioresistance and Increase the Efficacy of Radiotherapy: State of the Art

**DOI:** 10.3390/ijms231810211

**Published:** 2022-09-06

**Authors:** Fabio Busato, Badr El Khouzai, Maddalena Mognato

**Affiliations:** 1Department of Radiotherapy, Veneto Institute of Oncology IOV–IRCCS, 35128 Padua, Italy; 2Department of Biology, University of Padua, 35131 Padua, Italy

**Keywords:** radiobiology, radiation therapy, radioresistance, radiosensitizers

## Abstract

Cancer treatment with ionizing radiation (IR) is a well-established and effective clinical method to fight different types of tumors and is a palliative treatment to cure metastatic stages. Approximately half of all cancer patients undergo radiotherapy (RT) according to clinical protocols that employ two types of ionizing radiation: sparsely IR (i.e., X-rays) and densely IR (i.e., protons). Most cancer cells irradiated with therapeutic doses exhibit radio-induced cytotoxicity in terms of cell proliferation arrest and cell death by apoptosis. Nevertheless, despite the more tailored advances in RT protocols in the last few years, several tumors show a relatively high percentage of RT failure and tumor relapse due to their radioresistance. To counteract this extremely complex phenomenon and improve clinical protocols, several factors associated with radioresistance, of both a molecular and cellular nature, must be considered. Tumor genetics/epigenetics, tumor microenvironment, tumor metabolism, and the presence of non-malignant cells (i.e., fibroblast-associated cancer cells, macrophage-associated cancer cells, tumor-infiltrating lymphocytes, endothelial cells, cancer stem cells) are the main factors important in determining the tumor response to IR. Here, we attempt to provide an overview of how such factors can be taken advantage of in clinical strategies targeting radioresistant tumors.

## 1. Introduction

Radiotherapy (RT) causes the death of cancer cells mainly via radiation-induced DNA damage. Under normal conditions, the biological effects of ionizing radiation (IR) on mammalian cells involve the activation of the DNA-damage response (DDR), a complex pathway addressed to maintain genome integrity through the activation of proteins involved in sensing, signaling, and transducing the DNA damage signal to effector proteins of cell cycle progression/arrest, DNA repair, and apoptosis [1,2]. Different types of DNA damage can originate from the direct and indirect effects of IR. Base damage and single- and double-strand breaks (SSBs and DSBs) originate when IR hits the DNA molecule, which have been quantified, respectively, in 850 pyrimidine lesions, 450 purine lesions, 1000 SSBs and 20–40 DSBs/cell/Gy with γ-radiation in mammalian cells [3]. In addition to direct DNA damage, originating from the physical interaction between IR and the DNA helix, water radiolysis occurs in irradiated cells, and the reactive oxygen species (ROS) produced give rise to oxidative stress and DNA damage generation [4,5] that contribute to the cytotoxic effects of IR [6]. Oxidative DNA damage consists of oxidized bases such as 8-Oxo-7,8-dihydro-2′-deoxyguanosine (8-oxodGuo), oxidized pyrimidine derivatives (i.e., thymine glycol and 5,6 dihydrouracil), oxidized base-derived apurinic/apyrimidinic sites and SSBs [7]. According to the nature of IR-induced DNA damage, different pathways come into play to repair the lesion and restore cell functionality. In mammalian cells, both quiescent and proliferating, non-homologous end joining (NHEJ) is the predominant repair pathway for DSBs, in contrast to homologous recombination (HR), which intervenes only in cycling cells when sister chromatids are available in S-G_2_ phases [4]. Base excision repair (BER) and nucleotide excision repair (NER) are also active in repairing radio-induced DNA damage in the forms of single-strand breaks (SSBs) and base/nucleotide lesions [8]. Ultimately, irradiated cells can affect the physiology of non-directly irradiated cells due to the “bystander effect”, which is a cause of elevated levels of DNA damage measured as γ-H2AX foci, micronuclei, sister chromatid exchanges [9].

DDR pathway functionality is intimately associated with radioresistance in human cancer cells, and new approaches have been explored to counteract this phenomenon through the targeting of different pathways. Furthermore, IR and the subsequent DNA damage are a way to induce cancer cell senescence, together with replicative exhaustion, non-IR, genotoxic drugs, oxidative stress, and demethylating and acetylating agents. This is important because senescent cells are associated with tumor suppression. Recent data show that senescent cancer cells trigger strong antitumor protection mediated by antigen-presenting cells and CD8 T cells [10,11,12]. This response is superior to the protection elicited by cells undergoing immunogenic cell death. These preclinical data are expected to be more deeply investigated in order to applicate them in clinical practice.

Given the complexity of tumor radioresistance, in the following paragraphs, we make a brief overview dealing with the basic concepts of radiobiology and radiotherapy, the molecular/cellular characteristics of radioresistant cancer cells, and present and future targets against radioresistant tumors both in pre-clinical and clinical settings.

## 2. Molecular and Cellular Features Associated with Radioresistance

### 2.1. Basics of Radiobiology and Radiotherapy

The most common forms of IR used for cancer treatment in clinical practice are photons, electrons, and charged particles such as protons and carbon ions. The different types of radiation are administered by using specific radiation machines called LINAC (linear particle accelerators), eventually accelerated in a cyclotron or synchrotron in the case of charged particles, ultimately generating beams that reach tumor targets deep in the body. The different types of IR have different physical and biological properties that contribute differently to tumor mass and, therefore, are employed accordingly. Cellular and tissue effects induced by the radiation–cell interaction depend on the energy-deposition events of radiation, called linear energy transfer (LET), as well as the amount of intra- and extracellular water and the extracellular matrix (ECM). Low-LET radiation (such as γ- and X-rays) is called “sparsely ionizing” since it deposits its energy in all space directions, whereas high-LET radiation (such as protons and c-ions) is called “densely ionizing” since it deposits the energy along the primary track, causing more complex and clustered DNA damage [13,14,15]. Based on these differences, the relative biological effectiveness (RBE) of low- and high-LET radiation varies, being higher for the second one. RBE is defined as the ratio of doses to reach the same level of effect when comparing test radiation (i.e., protons or c-ions) and photons (X-rays or gamma rays) as the standard reference radiation [14,16]. In clinical practice, treatment planning considers RBE and LET to administer the best therapeutic doses to kill cancer cells and minimize the risk of adverse effects [17]. Difficulties arise when tumors are resistant to radiation treatment and obstruct the success of RT.

Radioresistance is defined as the capacity of cancer cells to survive and grow despite the deleterious DNA lesions induced by IR. Mechanisms that serve as the basis of radioresistance rely on the abnormal expression of key molecules, for example, proteins of the DNA-damage response (DDR) pathway, chromatin remodelers, and non-coding RNAs, as well as on the dysfunctional behavior of the cells present within the tumor environment. Tumor cell metabolism and dietary modifications also impact the response to RT [18]. Radioresistance can be intrinsic or acquired, the former being mainly associated with the molecular features of cancer cells, such as dysregulation in the DDR pathway; chronically activated proliferative, invasive, antiapoptotic signaling pathway; and IR-induced ROS level [5,6,19], the second being mainly associated with modifications in the cell compartment within the tumor environment such as enrichments in cancer stem cells (CSCs), which are epithelial to mesenchymal transition and the activation of transcription factors involved in CSC stemness maintenance [20,21]. Acquired radioresistance established following repeated IR fractions is also associated with the enhancement of DNA repair capacity and autophagic cell death [22,23]. Nevertheless, the distinction between intrinsic and acquired radioresistance is difficult to establish among the heterogeneity of human cancers, and we cannot exclude that both types can coexist within the same tumor.

DNA damage caused by IR and the inherent DNA repair capacity of tumor cells are important factors that determine therapeutic outcomes. RT causes the death of cancer cells mainly with IR-induced DSBs, which are the most important cytotoxic lesions induced by IR. Being actively and rapidly proliferating, cancer cells are affected by radio-induced DNA damage, which is why RT has a significant clinical impact on many tumors. Nevertheless, the dysfunctional expression of proteins, having a critical role in DNA damage signaling and repair, cell cycle progression, and apoptosis, is the origin of the malignant status of cancer cells and, in many cases, associated with a radioresistant phenotype [24]. IR-induced DSBs are the lesions that most contribute to the death of cancer cells, but they can be efficiently repaired either by HR or NHEJ. Deficiencies in the essential proteins of the NHEJ and HR pathways affect radiosensitivity in both normal and cancer cells, highlighting the central role of NHEJ in DSB repair [25,26,27]. Notably, the two DSB repair systems operate differently in cancer cells in response to proton- and photon-irradiation, indicating the preference of proton-induced DNA damage for HR in human cancer cells [27,28,29]. The reason is related to the different nature of the DNA damage, which is more complex and clustered after proton-irradiation [15] and can impede the activity of the Ku protein of the NHEJ pathway [30]. However, when both systems are active (i.e., in late S and G_2_ phases), NHEJ can operate when the lesion is not too complex; otherwise, there is a switch toward HR [27]. Elevated DNA repair capacity due to the overexpression of the DSB repair proteins RAD51 and DNA-PKcs confers radioresistance in different cancer cells [26,31,32,33,34]. The elevated expression of XRCC1, which is involved in the BER pathway and in SSB repair, is associated with radioresistance in patients with lung cancer and NSCLC [35]. Other HR and NHEJ proteins are dysregulated in radioresistant tumors, as well as key factors in the BER system, as finely reviewed by Carlos-Reyes et al. [36]. DSB repair kinetics, monitored by γ-H2AX immunofluorescence, was enhanced in radioresistant head and neck carcinoma cells [37]. Recently, Li et al. [38] reported that the radioresistance of esophageal carcinoma cells was associated with increased DNA repair by the NHEJ pathway due to high expression of the E3 ubiquitin–protein ligase RAD18. Cell cycle alterations are frequently observed in cancer cells and can influence their radiosensitivity. The overexpression of the cell-cycle-related protein p21Cip1/Waf1 renders cancer cells resistant to radiation [39,40]. P21 transcription is regulated by p53, the key transcription factor of the DDR pathway, which works together with p53-family proteins during the cell response to IR [41]. Alterations in the p53 pathway have a role in the radioresistance of pediatric tumors [42,43,44], endometrial cancer [45], head and neck cancer [46], rhabdomyosarcoma [47], and oral squamous cell carcinoma [48]. In addition to p53, the NK-kB family of transcription factors is also an essential mediator of tumor radioresistance through the modulation of hundreds of genes involved in cell survival, proliferation, angiogenesis, inflammation, and metastasis [49]. Radiation-induced apoptosis is the main mode of cell death, but dysfunctionality in the apoptotic program, such as the overexpression of antiapoptotic proteins and the downregulation of pro-apoptotic proteins, is the reason for apoptosis evasion in radioresistant cells. The expression of the anti-apoptotic protein XIAP has been increased in radioresistant human cell lines [50]. Considering that changes in protein expression levels are due to genomic alterations of both a genetic and epigenetic nature, we present below those that are related to radioresistant phenotypes by focusing on the DDR pathway.

### 2.2. Genetic and Epigenetic Features Associated with Radioresistance

#### 2.2.1. Genetic Variants

Genetic variants refer, in general, to changes in the nucleotide sequence, which manifest with different frequencies within a population. Rare variations that may or may not cause phenotypic changes are called mutations [51], whereas variations that occur in a population with a frequency ≥1% are called polymorphism [52]. Mutations in the *TP53* gene have been demonstrated to be associated with the radioresistance of pediatric brain tumors [43,46,53], and genetic polymorphism in the form of single nucleotide polymorphisms (SNPs) is associated with radioresistance. Variants of the *ATM* gene can impact cell radiosensitivity; indeed, patients with the *ATM* rs664677 TC variant have shown increased radiation resistance compared with the rs664677 TT genotype [54]. Qiu et al. [55], by using a 571 tumor-related gene panel for next-generation sequencing, identified the genetic variants that are associated with resistance to RT in cervical cancer patients. Among the genes of DDR, *BRCA1*, *BRCA2*, *ATM*, and *TP53* were mostly affected, and they showed a high frequency of copy number variations. In tumor samples from breast cancer patients, Bernichon et al. [56] identified genomic rearrangements in the form of mutations, amplifications, and deletions associated with radioresistance. They demonstrated that mutations in the *PIK3CA* gene, coding for the catalytic subunit p110 of the phosphatidylinositol 3′-kinase (Pl3K), are related to cell radiosensitivity. On the other hand, a relationship between RT efficacy and genetic polymorphism in the DNA repair genes *ERCC1/2* has been demonstrated in patients affected by NSCLC [57,58]. Gong et al. [59] demonstrated the association between RT efficacy and *rs*25487 polymorphism in the DNA repair gene *XRCC1* in patients affected by esophageal cancer. A potential association between *rs*9642880 G>T polymorphism in the *c-MYC* gene and the survival of patients affected by hepatocellular carcinoma treated with RT has also been reported [60].

#### 2.2.2. DNA Methylation Changes

DNA methylation is an epigenetic mechanism catalyzed by DNA methyltransferases, which controls gene expression by adding methyl groups to the C5 position of the cytosine in specific gene regions to repress transcription [61]. DNA methylation changes occur in cells irradiated with IR, and such changes are strictly related to specific genomic regions [62]. In general, it is thought that DNA methylation changes occur during the phenomenon of acquired radioresistance. An increased methylation level in the promoter region of the human telomerase reverse transcriptase gene (*hTERT*) causes its overexpression and contributes to the RT resistance of small cell lung cancer [63]. Methylation profiles of HNSCC patients identified an RT-related methylation signature in four genes (*ZNF10*, *TMPRSS12*, *ERGIC2*, *RNF215*), which was able to predict survival outcomes [64]. The methylation status of cathepsin E (*CTSE*) was inversely correlated with the radioresistance of patient-derived organoids [65]. The overexpression of DNA methyltransferase 3B (DNMT3B) following IR has been reported to be responsible for the radioresistance of nasopharyngeal carcinoma cells by suppressing the p53-mediated apoptotic pathway [66]. The levels of DNMT1 were found to be significantly increased in radioresistant head and neck cancer cell lines and associated with the promoter hypermethylation of the *PTEN* gene and its subsequent suppression [67].

#### 2.2.3. Histone Post-Translational Modifications

Radiation-induced alterations in histone post-translational modification (PTM) patterns can affect the chromatin structure necessary for the proper activation of DDR [68]. Several authors have demonstrated transient epigenetic changes following radiation, such as a decrease in histone phosphorylation and acetylation in response to DNA damage [69,70]. Epigenetic changes, intrinsic or developed during RT, whether transient or stable, can have short-term or long-term effects on gene expression and chromatin structure that contribute to rendering the cells more resistant to radiation. For instance, breast cancer cells become radioresistant when the enzymatic activities of histone acetyltransferase (HAT) and histone deacetylase (HDAC) are, respectively, low and high [71]. In prostate cancer cells, irradiation induces histone methylation at the promoter of the aldehyde dehydrogenase 1A1 (*ALDH1A1*) gene [72]. This protein belongs to the ALDH family, whose members are overexpressed in cancer stem cells in most carcinomas, such as prostate, skin, liver, brain, head and neck, lungs, and colorectal cancers [73]. The status of histone H3 lysine 27 trimethylation (H3K27me3) is associated with radiation response; indeed, Kim et al. [67] showed that high levels of H3K27me3 correlate with the radioresistance of head and neck cancer cell lines. On the contrary, in medulloblastoma patients, Gabriel et al. [74] showed that H3K27me3 deficiency correlates with radioresistant phenotypes because of an epigenetic shift from H3K27 trimethylation to acetylation, which activates the AKT pathway.

Interestingly, chromatin structure is directly associated with the functionality of DDR proteins, influencing the cell response to radio-induced DNA damage [75]. The level of chromatin condensation is higher in cells cultured in 3D compared to 2D conditions, and the higher amount of heterochromatin has a protective effect on radio-induced DNA damage [76].

#### 2.2.4. Transcriptomic Changes

Gene expression profiles associated with radioresistance have been reported for several cancer cells. The genetic signatures related to the radioresistance of head and neck squamous cell carcinoma (HNSCC) are associated with the high expression of genes responsible for epithelial–mesenchymal transition (EMT) and a seven-gene signature [77,78], as recently confirmed in [79]. Foy et al. [80] identified thirteen differentially expressed genes (including *CCND1)* that are significantly associated with the radioresistance of HNSCC cells and patients, and 12 of the 84 genes involved in DNA-damage signaling and repairing were under-expressed in radioresistant HNSCC cells after irradiation [37]. The gene networks of the DNA-damage response pathway (DNA repair, cell cycle, apoptosis), EMT pathway, chromatin organization pathway, cytokine production, stem cell differentiation pathway, and metal iron pathway are significantly associated with the radioresistance of nasopharyngeal carcinoma [81]. Many data are available for established radioresistant cell lines cultured in laboratories, but few recent studies have performed genome-wide analyses on tumor tissues (or organoids) derived from patients. In the study of Lee et al. [65], an organoid model of rectal cancer cells derived from patients was used to identify differentially expressed genes associated with radio-responsiveness. The authors were able to identify enriched pathways associated with radioresistance, which included DNA repair, immune response, and cell cycle progression pathways. These biological enrichments overlap with others from different studies, but with differences in individual differentially expressed genes, and this is the difficulty of detecting single genes responsible for or associated with radioresistance. 

Gene expression is modulated by the action of non-coding RNAs (ncRNAs), either long (>200 nt) or small (<200 nt), which function in the nucleus and in the cytoplasm [75,82,83,84,85]. Recent evidence has identified radioresistance-associated lncRNAs in NSCLC cells [32]. However, many studies on tumor radioresistance have reported changes in the expression of microRNAs (miRNAs), which are small, well-known ncRNAs that modulate the expression of many genes belonging to different pathways [86]. Thanks to genome-wide analyses carried out by microarray and RNA sequencing technologies, it is possible to identify differentially expressed miRNA species in radioresistant lung [87], breast [88,89], nasopharyngeal [90], prostate [91], rectal [65], head and neck, and cervical tumors [89].

Genome-wide analyses performed in the very last years with microarray and RNA sequencing technologies and integrated transcriptomic analyses using informatic analyses have improved the identification of genomic alterations. Recent studies have reported that transcriptional profiles can be useful in classifying subtypes in early advanced-stage lung adenocarcinoma and identifying better therapeutic strategies leveraging in vitro, in vivo, and clinical trial data [92]. In addition, transcriptome profiles can be associated with the histological subtypes of NSCLC [93]. However, the genetic and epigenetic scenario in radioresistant tumors is complicated to describe since, on the one hand, genetic and epigenetic signatures drive tumorigenesis, and on the other hand, RT itself can trigger long-term genetic and epigenetic changes in cancer cells that, in turn, affect radioresistance. Few studies have analyzed the role of ncRNAs in radioresistance and the metabolic phenotypes of radioresistant cancer cells [94,95]. While differentially expressed genes and ncRNAs following IR have been identified in numerous tumors, what is still little known is the relationship between specific genes/ncRNAs and radioresistance. The difficulty is due to the wide range of interconnected biological pathways that are either the basis of radioresistance or are activated in response to radiation. Additional studies employing a genome-wide approach, together with clinical treatments, will be required for future investigations on tumor radioresistance.

### 2.3. Tumor Microenvironment and Tumor Metabolism in Radioresistant Tumors

The tumor microenvironment (TME) is characterized by the presence of malignant and non-malignant cells such as cancer-associated fibroblasts (CAFs), tumor-associated macrophages (TAMs), tumor-infiltrating lymphocytes (TILs), and stromal cells (including dendritic cells, adipocytes, and endothelial cells) (Figure 1), all immersed in the extracellular matrix (ECM) [96]. CAFs are heterogeneous cells that originate from different types of cells, such as resident fibroblasts, bone marrow-derived mesenchymal cells, adipocytes, endothelial cells, and stellate cells [97,98]. CAFs are in a persistent state of activation to secrete components of ECM and immunomodulatory factors that support tumor growth [99]. CAFs are highly radioresistant cells because they undergo senescence rather than cell death following radio-induced DNA damage [100]. TAMs are phagocytic cells that are present in different forms and have different functions within the tumor mass. The M2-like macrophages support tumor growth by secreting molecules that stimulate angiogenesis and express immunosuppressive molecules [101,102]. Radioresistance has been observed following cell fusion between human M2 macrophages, and breast cancer cells form hybrid malignant clones resistant to RT [103]. High levels of M2 macrophages are found in radioresistant glioblastoma [104] and head and neck tumors [105]. TILs consist in lymphocytic cell populations composed of T lymphocytes (cytotoxic and helper) and B and NK cells, which can infiltrate the tumor environment and suppress the antitumor immunity, contributing to radioresistance [106]. The characteristics and function of CAFs, TAMs, and TILs, as well as their association with radioresistance, have been extensively described in recent reviews [107,108] and will not be further discussed here.

Most tumors, either solid or hematopoietic, contain populations of cells that exhibit stem cell properties: cancer stem cells manifest resistance to radio- and chemotherapy against the tumor and lead to its relapse. The best-known factors that render CSCs resistant to radiation are: (i) increased DNA repair capacity [109] due to delayed cell cycle progression and there being more time for efficient DSB repair by HR rather than NHEJ [110]; (ii) low levels of ROS thanks to ROS scavenger upregulation; and (iii) quiescence [111]. CSCs can be generated by the epithelial–mesenchymal transition (EMT) process [112], which is also associated with radioresistance. The number of CSCs within a tumor is proportionally linked to the success of therapy; indeed, tumors with few CSCs have a better prognosis compared with those with high levels of CSCs [113].

The metabolism of cancer cells is another important factor that dictates the efficiency of curative protocols, including RT. Cancer cells rely on anaerobic and aerobic glycolysis rather than oxidative phosphorylation [114], and for this reason, among others, they can survive with respect to normal cells. Under glycolytic metabolism, the amount of molecular oxygen (O_2_) within the tumor environment is reduced due to the disorganized nature of the vasculature [115], generating the condition of hypoxia, which comprises three different categories within the solid tumor mass: chronic hypoxia, acute hypoxia, and cycling (or intermittent) hypoxia [116,117]. Chronic hypoxia is when cells experience insufficient O_2_ amounts in a quasi-steady state, whereas acute and cycling hypoxias refer to temporal variations in pO_2_ [118]. Hypoxia is strictly associated with the resistance of cancer cells to IR-induced cytotoxicity, mainly because, with low levels of oxygen, the generation of ROS is reduced, and, consequently, DNA damage is lessened [115,119]. Conversely, well-oxygenated cells respond better to RT by a factor of 2.5–3 as a consequence of the oxygen enhancement ratio, which is an important component in photon and low-LET therapies [120]. The oxygen effect is most commonly explained by the oxygen fixation hypothesis, which postulates that radically induced DNA damage can be “fixed” by molecular oxygen, rendering DNA damage difficult or impossible for the cell to repair. Therefore, under hypoxic conditions, cancer cells become less sensitive to radio-induced DNA damage and resist to RT despite the activation of the ATM signaling pathway [121]. Interestingly, such pathways are activated under hypoxic conditions even in non-irradiated cells, suggesting that ATM activation can occur in the absence of DSBs [122]. Cancer cells can decrease endogenous ROS levels by activating multiple antioxidant enzymes (i.e., glutathione reductase, superoxide dismutase, thioredoxin reductase, catalase), which provide antioxidant molecules to protect themselves against oxidative stress and can be responsible for acquired radioresistance [6,123]. Under hypoxic conditions, the hypoxia-inducible factor (HIF-1) is stabilized to activate a transcriptional program that stimulates glycolysis and downregulates oxidative phosphorylation [124]. HIF-1 is constituted by the two subunits alpha and beta (HIF-1α and HIF-1β), whose dynamics vary within hypoxic cancer cells. HIF-1α stability increases during cyclic hypoxia compared with chronic hypoxia, whereas HIF-1β is insensitive to variations in oxygen levels, as recently found in Saxena and Jolly [116]. Once activated HIF-1 triggers a transcriptional pathway of hundreds of genes—including those of glucose metabolism, cell cycle regulation and proliferation, apoptosis, immune response, the protection of tumor blood vessels, and angiogenesis [94,125,126,127,128]—which ultimately allow for the cellular adaptive response to hypoxia [125]. This consists of the formation of new blood vessels [126], a shift to anaerobic metabolism for cellular energy production, increased apoptosis, and increased myeloid cell migration in inflamed areas [129].

Metabolic changes due to alterations in glucose and mitochondrial metabolic pathways contribute to the radioresistance of cancer cells [130,131]. Alterations in the expression levels of the key proteins and molecules of glycolytic processes (i.e., glucose transporters, lactic acid, pyruvate kinase, hexokinases) and mitochondrial functions (i.e., adenosine monophosphate family protein 3A, SIRT3, mitochondrial MAPK phosphatase) have been reported in radioresistant cancer cells [131]. A transient decrease in mitochondrial function has been observed in cancer cell lines shortly after exposure to IR, which correlates with the oxidizing effects of IR [132] and could be considered for tumor radioresistance. Indeed, colon and lung cancer cells that shift their metabolism to glycolysis because of mitochondria depletion manifest radioresistance [133,134]. On the other hand, an increase in mitochondrial abundance and a high oxidative metabolism have been reported in radioresistant vs. radiosensitive human HNSCC cells [135]. To complicate the picture of tumor cell heterogeneity, some cancer subtypes exhibit high oxidative phosphorylation, which can be considered a target for clinical application [136]. To add further complexity, hypoxia can impact IR-induced cytotoxicity in a radiation-quality-related manner. Carbon ions (high-LET radiation) are efficient in counteracting the radioresistance of HNSCC cells since they exert their action regardless of oxygen, in contrast to photons (low-LET radiation), whose action is dependent on oxygen concentration [137]. Interestingly, the acquisition of radioresistance in vitro in cancer cells can be exerted by repeated irradiation with photons (X-rays) but not with particles (c-ions) [20].

## 3. Targeting Strategies against Radioresistant Tumors

Efforts to improve the therapeutic ratio have led to the development of some agents that act to increase the radiosensitivity of cancer cells or to protect normal cells from the effects of radiation (Figure 2). A radiosensitizer is a drug that makes cancer cells more sensitive to radiation therapy. These compounds apparently promote the scavenging of free radicals produced by radiation damage on the molecular level. Radiation therapy generally affects DNA; mainly, it leads to DNA DSBs. Therefore, many radiosensitizing agents have been formulated to target the clinically developed DNA DSB repair pathways [138]. Other agents instead target different pathways, e.g., DNA-PKcs, ATM, and ATR signaling cascades. More than seven PARP inhibitors, for example, are currently being developed considering their role in DNA repair, especially for tumors with DNA repair defects, such as BRCA mutation, because of their synthetic lethality [139].

### 3.1. Hypoxic Cell Radiosensitizers

#### 3.1.1. Hyperbaric Oxygen

Hyperbaric oxygen (HBO) therapy is the inhalation of 100% oxygen at elevated pressure >1.5 atmospheres absolute (ATA; 150 kPa), typically 2–3 ATA (200–300 kPa). The physiological effects of HBO include short-term effects such as vasoconstriction and enhanced oxygen delivery, edema reduction, and phagocyte activation [140].

Most tumors contain oxygen-deprived compartments. The sterilization of hypoxic tumor cells requires a three-times higher radiation dose than cells with normal oxygen tension. HBO therapy is an effective approach to cope with the phenomenon of hypoxia by increasing the oxygen load of the tumor and, therefore, enhancing the response to irradiation [141]. Some studies have shown an increase in the 5-year survival rate of patients with cancers of the uterine cervix and head and neck.

#### 3.1.2. Carbogen

The idea of improving the oxygenation of tumors by breathing highly oxygenated air was recently revived by experiments in which participants breathe carbogen, a mixture of 95% oxygen and 5% carbon dioxide that does not produce the vasoconstriction associated with breathing 100% oxygen. Breathing carbon at atmospheric pressure is an attempt to overcome chronic (diffusion-limited) hypoxia by much simpler means than using hyperbaric chambers [142].

#### 3.1.3. Nicotinamide

Nicotinamide is an amide of vitamin B3. Acute hypoxia within tumors arises from the intermittent closure of blood vessels, resulting in fluctuations in the tumor’s microcirculation. Nicotinamide overcomes acute hypoxia by reducing these changes in the microcirculation. Furthermore, when nicotinamide is combined with carbogen, additional tumor sensitivity to radiation has been demonstrated, with overall enhancement ratios of between 1.8 and 2.1 in animal models using a clinically relevant dose schedule of 2 Gy day. It has, therefore, been proposed that the combination of carbogen and nicotinamide provides the optimal means of overcoming tumor hypoxia [143]. The cellular mechanism of action for a representative molecule from this class was found to be the G1 arrest, accompanied by the activation of p53/p21 DNA-damage signaling pathways, and most complexes induce high levels of apoptosis in low micromolar doses.

#### 3.1.4. Metronidazole and Its Analogs

Knowledge of the oxygen effect led to the development of compounds that mimic the radiosensitizing property of oxygen. The radiosensitizing abilities of hypoxic cell sensitizers have been observed to correlate with electron affinity [144]. Metronidazole and its analogs, such as misonidazole, etanidazole, and nimorazole, have been found to be effective in sensitizing hypoxic tumor cells [145].

The “oxygen fixation” hypothesis was proposed to explain the mechanism of action for this class of sensitizers [146]. They fix the radiation damage by preventing the chemical restitution of free radicals. Misonidazole has been observed to deplete sulfhydryl groups in cells, inhibit glycolysis, and repair potentially lethal radiation-induced cellular damage [147].

Nimorazole is a member of the same structural class as metronidazole but is less toxic, allowing for higher doses. As an oxygen mimetic, nimorazole induces free radical formation and is able to sensitize hypoxic cells to the cytotoxic effects of IR, thus preventing DNA repair and enhancing damage to DNA strands. A phase III study of nimorazole versus placebo in subjects with squamous cell carcinoma of the supraglottic larynx and pharynx demonstrated a statistically significant difference in improvement in the loco-regional control at 5 years post-treatment [148].

#### 3.1.5. Hypoxic Cell Cytotoxic Agents

Hypoxic cell cytotoxic agents include mitomycin-C and tirapazamine. Mitomycin-c is a bioreductive alkylating agent that has been studied in pancreatic, anal, and head and neck cancer. Tirapazamine is another bioreductive agent that is preferentially cytotoxic to hypoxic cells in vitro. It differs from oxygen-mimetic sensitizers, in that it requires metabolic activation and enhancement, as seen when this agent is given prior to or after RT [149]. Mitomycin-C, combined with radiation therapy, remains the standard-of-care therapy for anal carcinoma based on multiple clinical studies.

### 3.2. Radiosensitizing Chemotherapy Agents

#### 3.2.1. Fluoropyrimidines

5-fluorouracil (5-FU) and fluorodeoxyuridine (FdUrd) are analogs of uracil and deoxyuridine, respectively. They are thymidylate synthase (TS) inhibitors: interrupting the action of this enzyme blocks the synthesis of pyrimidine thymidylate (dTMP), which is a nucleotide required for DNA replication Randomized trials have demonstrated local control and survival advantages using systemic 5-FU and radiation compared with radiation alone in patients with rectal cancer, esophageal cancer, and pancreatic cancer [150]. 5-FU and FdUrd, through their metabolites, lead to cell cycle redistribution, DNA fragmentation, and cell death [151].

Capecitabine is an oral prodrug of 5-FU. It is converted to its cytotoxic form in three enzymatic steps, the last of which is mediated by thymidine phosphorylase. One of the potential advantages of this mechanism for increasing tumor cytotoxicity is that thymidine phosphorylase is overexpressed in tumor tissues. Interestingly, radiation has been shown to stimulate the expression of thymidine phosphorylase, which provides a further rationale [152].

#### 3.2.2. Gemcitabine

Gemcitabine is a nucleoside analog that mediates its antitumor effects by promoting apoptosis in malignant cells undergoing DNA synthesis [153]. It has demonstrated effectiveness as a single agent against solid tumors, including pancreatic cancer, non-small-cell lung cancer, head and neck cancer, and breast cancer. The mechanism by which gemcitabine radiosensitizes tumor cells is not yet clear. Preliminary studies indicate that the observed radiosensitization is not associated with either an increase in the radiation-induced DNA double-strand breaks or with a slowing of DNA double-strand break repair. This suggests that radiosensitization by gemcitabine is unlike that produced by fluoropyrimidines and thymidine analogs. The relationships between gemcitabine radiosensitization and DNA incorporation, alterations in DNA synthesis, and alteration in cell cycle kinetics remain to be investigated. In addition, it would be logical to investigate the role of apoptosis in gemcitabine-mediated radiosensitization, since this mechanism of cell death has been shown to be the pathway by which the drug exerts its cytotoxic action, at least in lymphoid cell lines [154].

#### 3.2.3. Taxanes

This group of anticancer agents works by disrupting microtubule function, thus inhibiting cell division [155]. Paclitaxel is the prototype of taxanes. Docetaxel is another agent in this group. Paclitaxel may be the most efficacious single chemotherapy agent for head and neck cancer, with a 40% response rate for patients with recurrent disease. As it is possible to achieve durable control with radiotherapy on locally advanced head and neck cancers in only a minority of cases; chemotherapy drugs, such as paclitaxel, are used with radiotherapy in an attempt to improve tumor control. Paclitaxel stabilizes microtubules and leads to the accumulation of cells in the G2/mitosis phase of the cell cycle, which is a necessary condition for its antitumor effect, and it is the phase with the greatest relative radiosensitivity. Paclitaxel has been shown to be a radiosensitizer in vitro for some, but not all, cell lines studied [156].

#### 3.2.4. Platinum-Based Drugs

This group of compounds, distinguished from most others by its metallic element base, has come to be recognized as one of the most potent chemotherapies available to date. Cisplatin (cis-diamminedichloroplatinum II), which is a prototype drug, has been acknowledged to be a potent radiosensitizer for many years and has had a significant role in clinical practice to date. Preclinical work performed using murine models by Rosenberg et al. in the late 1960s showed that cisplatin is an effective antitumor chemotherapy. Subsequent efforts have shown that its primary mechanism of inhibition for tumor growth appears to involve the inhibition of DNA synthesis. Another secondary mechanism includes the inhibition of transcription elongation by DNA interstrand cross-links [157]. Work on nonmammalian systems first demonstrated the radiosensitizing abilities of platinum-based compounds. This was confirmed in several mammalian systems as well [158] (Szumiel 1976).

This makes inherent sense because these platinum compounds have a high electron affinity and react preferentially with hydrated electrons. The exact mechanism of the increased cell death seen with combinations of IR and platinum drugs is not known for certain; however, the evidence would seem to point to the inhibition of PLDR49 and to the radiosensitization of hypoxic tumor cells [159]. Cisplatin-free-radical-mediated sensitization may involve the ability to scavenge free electrons formed by the interaction between radiation and DNA. The reduction in platinum moiety may serve to stabilize DNA damage that would otherwise be repairable.

Carboplatin, a second-generation platinum compound with a different toxicity profile, has also been studied as a radiosensitizer [160]. Its potential efficacy as a radiosensitizer has allowed for its incorporation into regimens used in several randomized trials. Interest exists in combining radiation with other platinum analogs, including oxaliplatin, as well as orally administered compounds such as satraplatin.

#### 3.2.5. Temozolomide

Temozolomide, a relatively new drug, is a second-generation alkylating agent, which is orally administered, is readily bioavailable, and demonstrates broad-spectrum activity in a variety of difficult-to-treat malignancies. It is unique in its ability to cross the blood–brain barrier (about 30% to 40% of plasma concentration found in CSF). Radiosensitization appears to occur via the inhibition of DNA repair, leading to an increase in mitotic catastrophe. It has proven efficacy as a first-line therapy for glioblastoma multiforme (GBM) patients in conjunction with RT, based on a randomized phase III clinical study demonstrating its survival benefit [161]. Temozolomide spontaneously converts into the reactive methylating agent MTIC and transfers methyl groups to DNA, the most important one being at the O6 position of guanine, an important site for DNA alkylation [162]. The MGMT gene encodes a DNA repair protein that removes the alkyl group from the O6 position of guanine, and high MGMT activity levels abrogate the effectiveness of alkylating agents. In vitro, temozolomide enhances the radiation response most effectively in MGMT-negative glioblastomas, likely due to decreased double-strand DNA repair capacity and increased DNA double-strand break damage, which occurs when a combination of temozolomide and radiation therapy is administered.

#### 3.2.6. Histone Deacetylase Inhibitors (HDACi)

The clinical effectiveness of histone deacetylase inhibitors (HDACis) as radiosensitizers has been demonstrated in vitro in cancer cells [163,164]. Even though HDACis have been validated in clinical trials and approved for cancer patients by the FDA for the treatment of cutaneous/peripheral T-cell lymphoma and multiple myeloma, many other HDACis are under investigation in clinical trials [165,166]. The evidence of HDACis being radiosensitizers in clinical practice is limited, and more experimentation is required to determine their potential [167].

#### 3.2.7. DNA Repair and Cell Cycle Inhibitors

Several drugs affecting DNA repair mechanisms and the progression of the cell cycle have demonstrated preclinical activity, and for this reason, they have promising clinical potential. Poly (ADP-ribose) polymerase (PARP) is a class of proteins with an essential role in DNA repair, detecting SSBs, recruiting DNA repair proteins, and ultimately stabilizing DNA [168]. The activity of PARP proteins is enhanced in many tumors; therefore, PARP inhibitors, which work by binding the SSB site and blocking the recruitment of repair proteins, are considered a promising strategy in combination with RT to enhance the efficacy of oncological cures [139]. We can consider the combination PARPi + RT feasible and usually safe, with hematological toxicities being the most commonly reported adverse events [169]. However, the clinical efficacy of this combination, as well as of other therapies such as immunotherapy, remains to be determined [170].

The PI3-kinase-like family of protein kinases includes DNA-PKcs (DNA-dependent protein kinase catalytic subunit), ATM (ataxia–telengiectasia mutated), and ATR (ataxia–telangiectasia and Rad3-related). This family recruits DNA repair proteins and activates cell cycle checkpoints in response to DSBs [171]. Silencing DNA-PKcs leads to increased radiosensitivity and DSBs [172,173]. Preclinical experiments have demonstrated that DNA-PKcs inhibitors increase the sensitivity of in vitro gastric cancer cells and that they can be effective and tolerable when associated with local RT [174,175].

The serine/threonine kinase ATM is activated by DNA DSBs to orchestrate the cellular response to IR. ATM inhibitors were studied in a phase I trial, which closed early due to a non-optimal pharmacokinetics profile [176]. Even though the development of that drug has been halted, the ATM pathway still represents an attractive therapeutic target, and second-generation ATM inhibitors are being investigated (ClinicalTrials.gov NCT04882917). Not only have these drugs been tested in monotherapy but also in combination. Indeed, the dual inhibition of DNA-PKcs and ATR represents a promising approach, concomitant with radiation [177]. Both ATR and its major downstream effector, checkpoint kinase 1 (CHK1) inhibitor, have been studied, and the results of phase I and II clinical trials have shown a low safety and efficacy profile, despite the promising preclinical studies [178].

Next-generation drugs with reduced toxicity and the possibility of selecting patients who benefit the most are future goals for this class of drugs.

### 3.3. Nanoparticles (NPs)

NPs usually have a simple structure composed of a core, a shell, and a surface. In the case of radiosensitizing NPs, the core is usually made of high-Z materials, such as silver, lanthanides, and (most extensively) gold, to exploit the increased photon absorption. The shell, which is chemically or physically bound to the core, acts as a base on which surface molecules (which sometimes include active agents) are anchored or bound with or without spacers. However, the high-Z elements can also be chelated by ligands present on the surface or inside the nanoparticle. The surface molecules usually consist of site-, tissue-, cell-, and/or receptor-specific molecules (targeting units) [179].

### 3.4. Immunomodulators

Immunotherapy has recently emerged as one of the major advances in prolonging overall survival in several cancers. It utilizes the patient’s immune system to induce tumor cell killing and can be either active or passive in nature. Active immunotherapy directly targets tumor cells and includes antibody therapy and chimeric antigen receptor T cell therapy. In contrast, passive immunotherapy enhances the ability of the immune system to eradicate tumor cells and includes immune checkpoint inhibitors (anti PD/PDL-1 and anti-CTLA-4) and cytokines [180]. The biological ways through which radiotherapy can stimulate the immune system and, thus, work synergically are: (i) by killing tumor cells and thus promoting the release of tumor antigens and the activation of cytotoxic T cells; (ii) by stimulating antigen-presenting cells [181]; (iii) by increasing MHC-1 expression [182]; and (iv) by releasing damage-associated molecular patterns (DAMPs) that can activate the immune system against tumor cells [183].

Several clinical trials have demonstrated the efficacy of radiotherapy combined with immune checkpoint inhibitors in real life: the PACIFIC trial and the Keynote 001 in NSCLC, Keynote-522 in triple-negative breast cancer, and many other practice-changing trials. Efforts are ongoing to better understand the detailed interaction between radiation therapy and the immune environment (sequencing, doses, timing) to increase their efficacy among cancer cures. One of these efforts includes making the abscopal effect more frequent in daily practice, as only 46 clinical cases using RT alone have been reported from 1969 to 2014 [184]. This is described as the ability of localized radiation to induce an antitumor response throughout the body in sites that were not subjected to targeted radiation [185]. Even though the abscopal effect was first described in 1953 [186], it has recently obtained great attention as a way to increase radiation therapy efficacy in combination with immune checkpoint inhibitors, as these drugs are revolutionizing cancer treatments and patient prognoses [187].

### 3.5. Radiation Therapy beyond Photons: Protons and Carbon Ions

Historically, radiation therapy has been associated with photons. Today, heavy ion accelerators, mostly carbon ions and proton ones, are being studied for their properties and, specifically, for their effects on radioresistant tumors. When compared to photons, carbon ions and protons show an inverted depth dose profile. Their energy deposition follows the Bragg curve, where low levels of energy are delivered to the normal tissue in the entrance channel and the maximum energy levels are delivered in the spread-out Bragg peak inside the tumor tissue, where the particles stop. Due to the steep energy drop after the Bragg peak, the normal tissue and the organs at risk beyond the tumor volume can be spared from radiation exposure [188]. Carbon ions also exhibit higher linear energy transfer (LET) than photons and protons [189]. This leads to a higher RBE, where damage caused by carbon ions is clustered in the DNA, overwhelming the cellular repair systems [190].

Thanks to its promising properties, heavy ion radiotherapy is being studied in several clinical trials, both for primary tumors (prostate cancer, bone cancer, sarcomas, and head and neck cancers) and recurrent tumors [191,192]. A review of head and neck cancers showed that, for malignant mucosal melanoma, the 5-year OS is higher with carbon ions than with photons (44% versus 25%), and for sino-nasal and paranasal cancers, the 5-year local control rate is higher with protons than with photons (88% versus 66%) [193].

In addition to its physical characteristics, high-LET radiation has been shown to induce complex DNA damage by inactivating hypoxic cells via direct ionization without the radiolysis of water [194]. Moreover, they perform increased immunogenicity in radiation-induced cell death compared to photon radiation through a variety of mechanisms, thus leading to a hypothesized advantage in the setting of combined immunotherapy [195] (Helm 2018). In mouse studies, carbon-ion irradiation correlates with stronger immune activation when paired with dendritic cell injection. Combining carbon-ion therapy with immunotherapy demonstrates increased antitumor immunity and reduces the number of metastases compared with RT or immunotherapy alone, or in combination with photons [196,197].

#### 3.5.1. SBRT

Conventional normo-fractionated radiotherapy (2 Gy/fraction) is ineffective for some tumors such as renal cell carcinoma (RCC), and they have been called “radioresistant” for this reason in the past few years. The recent use of SBRT (stereotactic body radiation therapy), thanks to the use of higher doses per fraction, has overcome some of these radioresistance scenarios. In the clinic, SBRT has been used to treat RCC, showing high local control and low toxicity rates. These data can be explained by the low α/β-ratio exhibited by RCC: low alpha–beta tumors are classically radioresistant to standard fractionation regimens and benefit from dose escalation using hypofractionation, which consists of delivering higher doses per fraction [40]. Using higher radiation doses, alternative cell death mechanisms, such as ceramide-induced apoptosis, have become more relevant in RCC cells [198]. Molecularly, a secretory form of acid sphingomyelinase is translocated to the extracellular leaflet of the cell membrane and transforms sphingomyelin into the pro-apoptotic protein ceramide via enzymatic hydrolysis [199]. The fact that acid sphingomyelinase, especially its secretory form, is predominantly expressed in endothelial cells explains the high sensitivity of endothelium to ceramide-induced apoptosis in RCC, a highly vascularized tumor. In vivo studies comparing sphingomyelinase-knockout mice with wildtype mice demonstrated that sphingomyelinase-knockout mice exhibited an increased threshold to irradiation-induced endothelial apoptosis and were resistant to single-dose RT with 20 Gy. The importance of sphingomyelinase activity regarding tumor response after SBRT was further underlined in the study by Sathishkumar et al.: 75% of the patients with partial or complete tumor response after SBRT exhibited significantly increased serum ceramide and serum sphingomyelinase levels, whereas none of the non-responders had increased levels of these proteins [200]. A systematic review showed that ablative SBRT can be effectively used to treat RCC with high local control rates (84–100%) [201].

#### 3.5.2. FLASH Radiotherapy

Favaudon et al. [202] discovered that pulsed and ultrahigh-dose-rate irradiation (≥40 Gy/s, FLASH) causes less damage to the healthy lung than conventional radiotherapy (≤0.03 Gy/s, CONV) in mouse models while preserving efficacy against tumor cells. They called this technology FLASH radiotherapy, and it has two major advantages: a low toxicity rate in irradiated healthy tissues, thus providing a chance to increase the dose to tumor targets, and a short delivery time; e.g., the first patient affected by cutaneous T-cell lymphoma was irradiated in 90 ms [203]. As shown in Figure 2, FLASH RT represents a new and promising field to fight cancer. Higher doses in the tumor target will hopefully increase efficacy for radioresistant tumors. Once we have better knowledge of its standard dosimetry, 3D treatment planning, volumetric image guidance, and motion management, more clinical trials will start to enroll patients [204].

### 3.6. Diet

Short-term fasting and calorie restriction have been associated with the mechanisms of radioresistance. Klement et al. [18] summarized them according to the 5Rs: (a) DNA Repair: short-term fasting likely selectively improves DSB repair in normal cells but not cancer cells (mTOR inhibition), thus favoring normal tissue repair and cancer cell death; (b) Repopulation (cell proliferation occurring during the course of fractionated RT in both tumors and normal tissue): calories restriction in rodents reduces IGF-1/insulin–PI3K–Akt–mTor signaling, which has been shown to be correlated with significant tumor growth delay [205]; (c) Redistribution: fasting seems to promote cell cycle progression, M phase accumulation, and energy expenditure, and in this way, it renders such cells synthetically vulnerable to the combination of nutrient restriction with RT or chemotherapy [206]; (d) Reoxygenation: calorie restriction downregulates VEGF [207], thus decreasing areas of hypoxia in tumors [208].

Though there is a large amount of preclinical data, further clinical data are necessary to establish the effects of calorie restriction and intermittent fasting on irradiated cancer cells. In accordance, recent ASCO guidelines state that, currently, there is insufficient evidence to recommend for or against dietary interventions such as ketogenic or low-carbohydrate diets, low-fat diets, functional foods, or fasting to improve outcomes related to QoL, treatment toxicity, or cancer control.

## 4. Conclusions

The complexity of tumor radioresistance relies on many different intrinsic and extrinsic factors that cannot be considered independently from each other. This is the major obstacle in developing new clinical RT protocols that can be efficient for the heterogeneity of oncologic patients. Mechanisms of radioresistance are only partially comprehended and need to be further studied and validated to develop clinical guidelines.

## Figures and Tables

**Figure 1 ijms-23-10211-f001:**
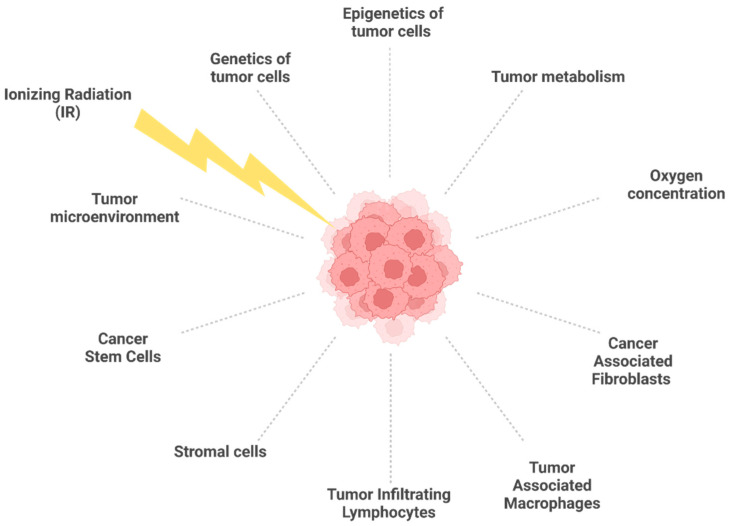
Biological factors affecting tumor radioresistance. In response to IR, the diverse components of the tumor microenvironment interact each other, with tumor cells contributing to radioresistance. Created with BioRender.

**Figure 2 ijms-23-10211-f002:**
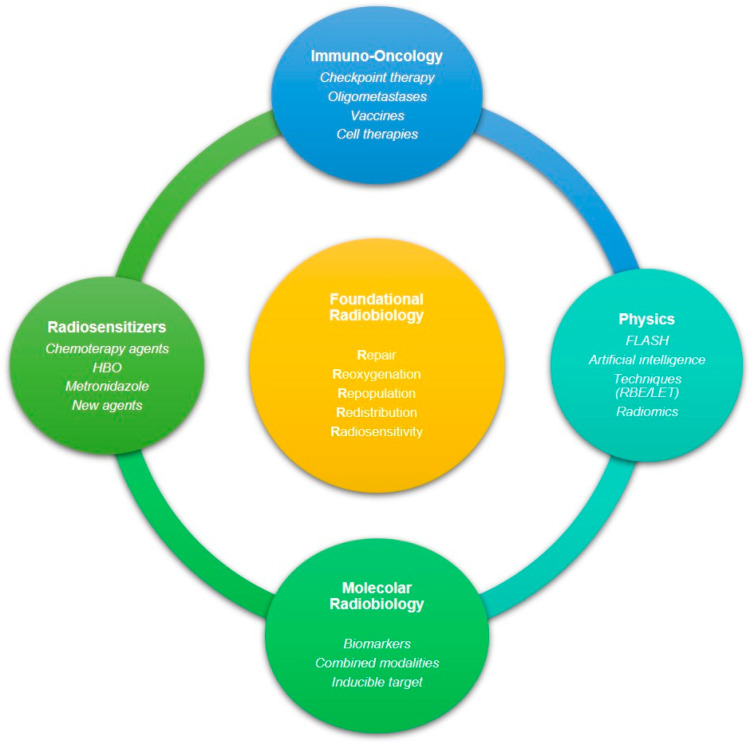
The multifaceted spectrum of radiation biology research, mechanisms, and clinical applications. In the center, the 5Rs explain the biological basis of radiobiology.

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
