# Peer review of "Biological Mechanisms to Reduce Radioresistance and Increase the Efficacy of Radiotherapy: State of the Art"

_ijms, 2022, doi:10.3390/ijms231810211_

Round 1

Reviewer 1 Report

In their article, Busato and colleagues present a review of the biological mechanisms and strategies for radioresistance and -sensitization of tumors. Radiosensitization of tumors is a clinically very relevant issue to improve treatment outcomes and prevent tumor relapse. A review of current developments would be of high interest to the broad field of oncology. However, the article has some shortcomings in terms of structuring and depth of coverage of the topic. It will be helpful if the authors can consider these points to improve their manuscript:

·          Line 24: Here and in the entire manuscript, the authors focus primarily on apoptosis as the mechanism of cancer cell death. However, most rapidly proliferating tumor cells, often with dysfunctional cell cycle checkpoints, undergo reproductive cell death by mitotic catastrophe, which renders them sensitive to IR as stated by the authors in lines 86/87. Another type of inactivation of the clonogenic potential of cancer cells is senescence. The Induction of senescence and senolysis as a potential radiosensitizer has not been considered in the entire manuscript, only briefly for CAFs but not for tumor cells.

·         Line 34: radio-induced or radiation-induced? I recommend using ‘IR-induced’ since this acronym for ionizing radiation has been introduced.

·         Line 54-58:

a   a) Why do the authors refer to X-rays (artificially produced) when they write about radionuclides that emit gamma-rays? Co60 indeed produces high-energy photons (> 1 MeV) used for radiotherapy, but not Cs which is not applied to treat tumors in deeper tissue layers, if at all. The most common source of high-energy photons used today for radiotherapy of tumor patients are medical linear accelerators.

b) Neutrons are not charged particles. Also, neutrons are super rarely used in radiotherapy as high-energy beams or in boron neutron capture therapy.

c) Particles for therapeutic purposes are usually accelerated in a cyclotron or synchrotron to achieve sufficient energy to reach deeper situated tumors but not only in a linear accelerator, although it may be a component of the accelerator system used for hadron therapy for initial acceleration between the source and synchrotron.

·         Line 57: Different radiation qualities have not only different physical but also biological properties, especially c-ions versus photons and protons. This is not mentioned until line 503 and following, but should be considered here already.

·         Line 63: Do the authors mean iron or ion? Iron ions are not relevant for radiotherapy but rather for cosmic rays. Better use: (protons, c-ions).

·         Line 84: IR has been introduced in the abstract and has been used already in line 28. Check the manuscript thoroughly for the introduction and use of acronyms, e.g., DSB in line 320, RBE in line 510….

·         Line 165 and the following: What about the application of histone deacetylases as radiosensitizers? This strategy should be discussed here.

·         Line 278: This effect has been attributed to the oxygen fixation hypothesis where molecular oxygen is needed to fix the DNA damage induced by indirect ionization causing water radiolysis. The authors should cite and explain this effect and introduce the oxygen enhancement ratio (OER) here since they pay very much attention to the effects of tumor hypoxia on radiotherapy. Only in line 366, it is briefly mentioned.

·         Line 301: The adaptive response should be briefly explained to the unprimed reader.

·         Line 321-322: Here and elsewhere, the authors place much emphasis on the impact and relevance of radiation-induced DNA damage. But the use of small molecule inhibitors targeting, e.g., DNAPKcs, ATM, and ATR signaling cascades or PARP to compromise DNA repair or the induction of cell cycle checkpoints by Chk1, Chk2, or Wee1 inhibitors is not addressed; or the possibility of induction of HR deficiency, e.g., by inhibitors of mTOR, tyrosine kinases, and others. Instead, the authors largely address the well-known additive or synergistic effects of radiochemotherapy with various cytostatic agents. Most of the radiosensitization strategies presented are not really ‘new’ (line 313).

     In general, the manuscript deals predominantly with hypoxia and cytostatics, and many other relevant and more current strategies for radiosensitization of tumors are only briefly mentioned and inadequately or not addressed.  Many relevant factors, some of which are touched in the introductory sections 1 and 2, are not subsequently discussed as targets for radiosensitization, e.g., targeting of IAPs (apoptosis), tyrosine kinases (EGFR and others), anaerobic glycolysis/Warburg effect (Glut1), telomerase inhibitors, vasculogenesis, DNA-repair, cell cycle checkpoints....

·         Lines 327-384: This extensive section deals with the effect of hypoxia on tumor radioresistance and should be grouped under the heading 'Hypoxia'.

·         Figure 2: Many of the approaches/radiosensitizers mentioned in Figure 2 are not explained or mentioned at all in the manuscript or the figure legend, such as FLASH radiotherapy (for dose escalation due to reduced normal tissue toxicities by oxygen depletion), inducible targets, and others…

·         Line 400: Please provide a reference.

·         Line 454: Oxaliplatin75?

·         Line 484: Besides a local effect, an important feature of immuno oncology/immune checkpoint inhibitors is the increase in efficacy in eradicating distant metastases as the abscopal effect. Since it represents an important radiosensitizing effect on metastases and thus treatment outcome, it should be mentioned by the authors.

·         Line 507: For particles, the depth dose profile is not sufficiently presented and described: Compared to photons, they show an inverted depth dose profile. Their energy deposition follows the Bragg curve, where low levels of energy are delivered to the normal tissue in the entrance channel, and the maximum energy in the (spread-out) Bragg peak in the tumor tissue when the particle stops. Due to the steep energy drop after the Bragg peak, the normal tissue and the organs at risk behind the tumor volume can be spared from radiation exposure.

·         Line 519: The induction of complex DNA damage and the ability to inactivate hypoxic cells by direct ionization without radiolysis of water should be mentioned here as the primary and elementary biological effects of c-ions followed by the immunogenic effects.

·         Line 534: The role and importance of the alpha/beta ratio for the radioresistance of tumor tissue must be explained to the reader. This important parameter, which defines the radiation response of a tissue, is introduced too abruptly here. Otherwise, the reader will not understand the relationship between a low alpha/beta ratio and radioresistance (large shoulder in clonogenic cell survival curves). Alternatively, it could be omitted here and RCC may be simply described as radioresistant.

·         Line 554: The 5R's of radiotherapy should be introduced much earlier in the manuscript as they define tumor response to radiotherapy and are the essential concept in radiation oncology and pertain to all the issues of radioresistance the authors address in the manuscript. Previously, they appear in Figure 2, without being referenced and without being explained further.

Author Response

Responses to Reviewer 1

In their article, Busato and colleagues present a review of the biological mechanisms and strategies for radioresistance and -sensitization of tumors. Radiosensitization of tumors is a clinically very relevant issue to improve treatment outcomes and prevent tumor relapse. A review of current developments would be of high interest to the broad field of oncology. However, the article has some shortcomings in terms of structuring and depth of coverage of the topic. It will be helpful if the authors can consider these points to improve their manuscript:

1.Reviewer’s comment:

Line 24: Here and in the entire manuscript, the authors focus primarily on apoptosis as the mechanism of cancer cell death. However, most rapidly proliferating tumor cells, often with dysfunctional cell cycle checkpoints, undergo reproductive cell death by mitotic catastrophe, which renders them sensitive to IR as stated by the authors in lines 86/87. Another type of inactivation of the clonogenic potential of cancer cells is senescence. The Induction of senescence and senolysis as a potential radiosensitizer has not been considered in the entire manuscript, only briefly for CAFs but not for tumor cells.

Authors’ response: We didn’t discuss about senescence and senolysis because these are not strict “radiosensitizers” as they do not make IR more effective but, indeed, could be seen as a consequence of IR in order to promote immunogenic response against cancer cells. Somehow, considering the interesting intrinsic biological connection with IR, we added a specific paragraph. Thank you.

Line 50 of the revised manuscript, we add as follows: “IR and the subsequent DNA damage is a way to induce cancer cell senescence, together with replicative exhaustion, non-ionizing radiation, genotoxic drugs, oxidative stress and demethylating and acetylating agents. This is important because senescent cells are associated with tumour suppression. Recent data show that cancer senescent cells trigger strong anti-tumor protection mediated by antigen-presenting cells and CD8 T cells. This response is superior to the protection elicited by cells undergoing immunogenic cell death. These preclinical data are expected to be more deeply investigated in order to applicate them in clinical practice”.

2.Reviewer’s comment:

Line 34: radio-induced or radiation-induced? I recommend using ‘IR-induced’ since this acronym for ionizing radiation has been introduced.

Authors’ response: we corrected.

3.Reviewer’s comment:

Line 54-58:

  1. a) Why do the authors refer to X-rays (artificially produced) when they write about radionuclides that emit gamma-rays? Co60 indeed produces high-energy photons (> 1 MeV) used for radiotherapy, but not Cs which is not applied to treat tumors in deeper tissue layers, if at all. The most common source of high-energy photons used today for radiotherapy of tumor patients are medical linear accelerators.
  2. b) Neutrons are not charged particles. Also, neutrons are super rarely used in radiotherapy as high-energy beams or in boron neutron capture therapy.
  3. c) Particles for therapeutic purposes are usually accelerated in a cyclotron or synchrotron to achieve sufficient energy to reach deeper situated tumors but not only in a linear accelerator, although it may be a component of the accelerator system used for hadron therapy for initial acceleration between the source and synchrotron.

Authors’ response: We thank the reviewer for pointing out this question. We edited as follows:” The most common form of IR used for cancer treatment in clinical practice are photons, electrons and charged particles like protons and carbon ions. The different types of radiation are administered by using specific radiation machines called LINAC (Linear Particle Accelerators), eventually accelerated in a cyclotron or synchrotron in case of charged particles, ultimately generating beams that reach tumour targets deep in the body.”

4.Reviewer’s comment:

Line 57: Different radiation qualities have not only different physical but also biological properties, especially c-ions versus photons and protons. This is not mentioned until line 503 and following but should be considered here already.

Authors’ response: Following the reviewer’s suggestion we edited as follows:” The different types of IR have different physical and biological properties that contribute differently on tumour mass and therefore are employed accordingly”. We prefer considering this point as it is for clarity.

5.Reviewer’s comment:

Line 63: Do the authors mean iron or ion? Iron ions are not relevant for radiotherapy but rather for cosmic rays. Better use: (protons, c-ions).

Authors’ response: we corrected.

6.Reviewer’s comment:

Line 84: IR has been introduced in the abstract and has been used already in line 28. Check the manuscript thoroughly for the introduction and use of acronyms, e.g., DSB in line 320, RBE in line 510….

Authors’ response: we corrected.

7.Reviewer’s comment:

Line 165 and the following: What about the application of histone deacetylases as radiosensitizers? This strategy should be discussed here.

Authors’ response: we welcome the reviewer’s suggestion. Since part 2 is dedicated to the description of molecular and cellular basis of radioresistance we added the new part in section 3, which is dedicated to discuss strategies overcoming radioresistance.

Line 577 of the revised manuscript, we added the following new paragraph entitled: “3.2.6 Histone deacetylase inhibitors (HDACi)”. “The clinical effectiveness of histone deacetylase inhibitors (HDACi) as radiosensitizers has been demonstrated in vitro in cancer cells [Camphausen and Philip J. Tofilon, 2007; Johnson et al., 2021]. Even though HDACi have already been validated in clinical trials and approved in cancer patients by the FDA for the treatment of cutaneous/peripheral T-cell lymphoma and multiple myeloma, many other HDACi are under investigation in clinical trials [Cappellacci et al., 2020; Bondarev et al., 2021]. Evidence of HDACi as radiosensitizers in clinical practice is limited and more experimentation is required to determine their potential [Antrobus et al., 2022].

8.Reviewer’s comment:

Line 278: This effect has been attributed to the oxygen fixation hypothesis where molecular oxygen is needed to fix the DNA damage induced by indirect ionization causing water radiolysis. The authors should cite and explain this effect and introduce the oxygen enhancement ratio (OER) here since they pay very much attention to the effects of tumor hypoxia on radiotherapy. Only in line 366, it is briefly mentioned.

Authors’ response: We understand the need to explain in depth the oxygen fixation hypothesis and for this reason we would add:

Line 321 of the revised manuscript “To quantify, well-oxygenated cells respond better to radiotherapy by a factor 2.5–3. This increased radio-response is known as the oxygen enhancement ratio. The oxygen effect is most commonly explained by the oxygen fixation hypothesis, which postulates that radical-induced DNA damage can be permanently ‘fixed’ by molecular oxygen, rendering DNA damage irreparable. [Grimes and Partridge, 2015]”.

9.Reviewer’s comment:

Line 301: The adaptive response should be briefly explained to the unprimed reader.

Authors’ response: Thank you, as abundant data are available, we thought to omit this; however, in order to make it easier to understand, we thank you for the advice and modify the sentence as follows:

Line 339 of the revised manuscript: “Once activated HIF-1 triggers a transcriptional pathway of hundreds of genes, including those of glucose metabolism, cell cycle regulation and proliferation, apoptosis, immune response, protection of tumor blood vessel and angiogenesis [Harada et al., 2016; Carmeliet et al., 1998; Jin et al., 2022; Wu et al., 2022], that ultimately allow the cellular adaptive response to hypoxia [102]. This consists of the formation of new blood vessels [103], the shift to anaerobic metabolism for cellular energy production, increased apoptosis and increased myeloid cells migration in inflamed areas” [Ziello et al., 2007].

10.Reviewer’s comment:

Line 321-322: Here and elsewhere, the authors place much emphasis on the impact and relevance of radiation-induced DNA damage. But the use of small molecule inhibitors targeting, e.g., DNAPKcs, ATM, and ATR signaling cascades or PARP to compromise DNA repair or the induction of cell cycle checkpoints by Chk1, Chk2, or Wee1 inhibitors is not addressed; or the possibility of induction of HR deficiency, e.g., by inhibitors of mTOR, tyrosine kinases, and others. Instead, the authors largely address the well-known additive or synergistic effects of radiochemotherapy with various cytostatic agents. Most of the radiosensitization strategies presented are not really ‘new’ (line 313).

Authors’ response: We welcome reviewer’s suggestion and we added as follows:

Line 418 of the revised manuscript: “Other agents instead target different pathways e.g., DNA-PKcs, ATM, ATR signaling cascades. More than seven PARP inhibitors, for example, have been currently developed considering their role of in DNA repair, especially for tumors with DNA repair defects, such as BRCA mutation, because of their synthetic lethality [Lesueur et al., 2017].

Moreover, we deleted “new” from the heading of section 3.

In general, the manuscript deals predominantly with hypoxia and cytostatics, and many other relevant and more current strategies for radiosensitization of tumors are only briefly mentioned and inadequately or not addressed.  Many relevant factors, some of which are touched in the introductory sections 1 and 2, are not subsequently discussed as targets for radiosensitization, e.g., targeting of IAPs (apoptosis), tyrosine kinases (EGFR and others), anaerobic glycolysis/Warburg effect (Glut1), telomerase inhibitors, vasculogenesis, DNA-repair, cell cycle checkpoints....

11.Reviewer’s comment:

Lines 327-384: This extensive section deals with the effect of hypoxia on tumor radioresistance and should be grouped under the heading 'Hypoxia'.

Authors’ response: We welcome the reviewer’s suggestion. We grouped the section 3.1. under the heading “Hypoxic cell radiosensitizers”.

12.Reviewer’s comment:

Figure 2: Many of the approaches/radiosensitizers mentioned in Figure 2 are not explained or mentioned at all in the manuscript or the figure legend, such as FLASH radiotherapy (for dose escalation due to reduced normal tissue toxicities by oxygen depletion), inducible targets, and others…

Authors’ response: We did not include FLASH RT because it is a novel technique, not a novel way to make tumours less radioresistant (at least not yet demonstrated).

Somehow, considering the expectations associated with it in the upcoming years, we will add a paragraph concerning FLASH RT. Thank you.

3.5.2 FLASH radiotherapy

 Flavudon et al (Favaudon V, Caplier L, Monceau V, Pouzoulet F, Sayarath M, Fouillade C, et al. Ultrahigh dose-rate FLASH irradiation increases the differential response between normal and tumor tissue in mice. Sci Transl Med. (2014) 6:245ra293. doi: 10.1126/scitranslmed.3008973) discovered that pulsed and ultrahigh dose-rate irradiation (≥40 Gy/s, FLASH) causes less damage to the healthy lung than conventional radiotherapy (≤0.03 Gy/s, CONV) in mouse models while preserving efficacy against tumour cells. They called this technology FLASH radiotherapy and it has 2 major advantages: low toxicity rate to irradiated healthy tissues thus giving the chance to increase the dose to tumour targets; Short delivery time, e.g. the first patient, affected by T-cell cutaneous lymphoma, was irradiated in 90 ms (Lin B, Gao F, Yang Y, Wu D, Zhang Y, Feng G, Dai T, Du X. FLASH Radiotherapy: History and Future. Front Oncol. 2021 May 25;11:644400. doi: 10.3389/fonc.2021.644400. PMID: 34113566; PMCID: PMC8185194.).

FLASH radiotherapy represents a new promising field to fight cancer. Higher doses to the tumour target will hopefully increase efficacy for radioresistant tumours. Once we’ll have a better knowledge of its standard dosimetry, 3D treatment planning, volumetric image guidance and motion management, more clinical trials will start to enrol patients (Taylor PA, Moran JM, Jaffray DA, Buchsbaum JC. A roadmap to clinical trials for FLASH. Med Phys. 2022 Jun;49(6):4099-4108. doi: 10.1002/mp.15623. Epub 2022 Apr 25. PMID: 35366339.).

13.Reviewer’s comment:

Line 400: Please provide a reference.

Authors’ response: We added the following reference:” Hasegawa K, Okamoto H, Kawamura K, Kato R, Kobayashi Y, Sekiya T, Udagawa Y. The effect of chemotherapy or radiotherapy on thymidine phosphorylase and dihydropyrimidine dehydrogenase expression in cancer of the uterine cervix. Eur J Obstet Gynecol Reprod Biol. 2012 Jul;163(1):67-70. doi: 10.1016/j.ejogrb.2012.03.014.

14.Reviewer’s comment:

Line 454: Oxaliplatin75?

Authors’ response: We corrected, thank you

  1. Reviewer’s comment:

Line 484: Besides a local effect, an important feature of immuno oncology/immune checkpoint inhibitors is the increase in efficacy in eradicating distant metastases as the abscopal effect. Since it represents an important radiosensitizing effect on metastases and thus treatment outcome, it should be mentioned by the authors.

Authors’ response: The abscopal effect was indirectly described in line 499, we made it more explicit. Thank you.

Line 613 of the revised version: “One of these efforts include making the abscopal effect more frequent in daily practice as only 46 clinical cases due to RT alone have been reported from 1969 to 2014 (Abuodeh Y, Venkat P, Kim S. Systematic review of case reports on the abscopal effect. Curr Probl Cancer. 2016;40:25–37). This is described as the ability of localised radiation to induce an antitumor response throughout the body at sites that were not subjected to targeted radiation (Craig DJ, Nanavaty NS, Devanaboyina M, Stanbery L, Hamouda D, Edelman G, Dworkin L, Nemunaitis JJ. The abscopal effect of radiation therapy. Future Oncol. 2021 May;17(13):1683-1694. doi: 10.2217/fon-2020-0994. Epub 2021 Mar 17. PMID: 33726502.).

Even though the abscopal effect was first described in 1953 (Mole RH. Whole body irradiation; radiobiology or medicine? Br. J. Radiol. 26(305), 234–241 (1953).Crossref, Medline, CAS, Google Scholar), it has recently obtained great attention as a way to increase radiation therapy efficacy in combination with immune check-point inhibitors as these drugs are revolutionising cancer treatments and patients’ prognosis” (Dagoglu N, Karaman S, Caglar HB, Oral EN. Abscopal Effect of Radiotherapy in the Immunotherapy Era: Systematic Review of Reported Cases. Cureus. 2019 Feb 20;11(2):e4103. doi: 10.7759/cureus.4103. PMID: 31057997; PMCID: PMC6476623.)

  1. Reviewer’s comment:

Line 507: For particles, the depth dose profile is not sufficiently presented and described: Compared to photons, they show an inverted depth dose profile. Their energy deposition follows the Bragg curve, where low levels of energy are delivered to the normal tissue in the entrance channel, and the maximum energy in the (spread-out) Bragg peak in the tumor tissue when the particle stops. Due to the steep energy drop after the Bragg peak, the normal tissue and the organs at risk behind the tumor volume can be spared from radiation exposure.

Authors’ response: ok, we made it clearer. Thank you.

Line 630 of the revised manuscript: “When compared to photons, carbon ions and protons show an inverted depth dose profile. Their energy deposition follows the Bragg curve where low levels of energy are delivered to the normal tissue in the entrance channel and the maximum energy levels are delivered in the (spread-out) Bragg peak inside the tumour tissue when the particle stops. Due to the steep energy drop after the Bragg peak, the normal tissue, and the organs at risk behind the tumour volume can be spared from radiation exposure [Blanchard et al., 2018]. Carbon ions also exhibit…”.

  1. Reviewer’s comment:

Line 519: The induction of complex DNA damage and the ability to inactivate hypoxic cells by direct ionization without radiolysis of water should be mentioned here as the primary and elementary biological effects of c-ions followed by the immunogenic effects.

Authors’ response: Ok, thank you, we made it clearer.

Line 647 of the revised manuscript: “In addition to its physical characteristics, high-LET radiation has been shown to induce complex DNA damage by inactivating hypoxic cells by direct ionisation without radiolysis of water (Schlaff CD, Krauze A, Belard A, O'Connell JJ, Camphausen KA. Bringing the heavy: carbon ion therapy in the radiobiological and clinical context. Radiat Oncol. 2014 Mar 28;9(1):88. doi: 10.1186/1748-717X-9-88. PMID: 24679134; PMCID: PMC4002206.). Moreover they perform increased immunogenicity of radiation-induced cell death compared to photon radiation through a variety of mechanisms…”

  1. Reviewer’s comment:

Line 534: The role and importance of the alpha/beta ratio for the radioresistance of tumor tissue must be explained to the reader. This important parameter, which defines the radiation response of a tissue, is introduced too abruptly here. Otherwise, the reader will not understand the relationship between a low alpha/beta ratio and radioresistance (large shoulder in clonogenic cell survival curves). Alternatively, it could be omitted here and RCC may be simply described as radioresistant.

Authors’ response: We hope the alpha/beta background is shared by the readers. Somehow, as it may not be the case we’ll add a line. Thank you.

Line 669 of the revised manuscript: “..by RCC: low alpha–beta tumours are classically radioresistant to standard fractionation regimens, and benefit from dose escalation using hypofractionation, that consists of delivernig higher doses per fraction (Nguyen EK, Quan K, Parpia S, Tran S, Swaminath A. Stereotactic body radiotherapy for osseous low alpha-beta resistant metastases for pain relief-SOLAR-P. Radiat Oncol. 2021 Sep 3;16(1):170. doi: 10.1186/s13014-021-01897-0. PMID: 34479581; PMCID: PMC8417953.).

  1. Reviewer’s comment:

Line 554: The 5R's of radiotherapy should be introduced much earlier in the manuscript as they define tumor response to radiotherapy and are the essential concept in radiation oncology and pertain to all the issues of radioresistance the authors address in the manuscript. Previously, they appear in Figure 2, without being referenced and without being explained further.

Authors’ response: We welcome the reviewer’s suggestion by adding in paragraph 2, line 84, as follows: “Tumor cell metabolism and dietary modifications also impact on the response to radiotherapy [Klement 2014]. If the reviewer agrees, we prefer to discuss only the association of 5R’s and diet in paragraph 3.6. We also edited the legend of Figure 2 as follows: “…and clinical application. In the centre the 5Rs explaining the biological basis of radiobiology”.

Reviewer 2 Report

The manuscript entitled “Biological mechanisms to reduce radioresistance and increase 2

the efficacy of radiotherapy: state of the art” discusses an important and relevant aspect of 

basic concepts in radiobiology and molecular characteristics of radioresistant cancer cells. 

However, although being important, the authors only mention important effects of hypoxia and metabolism-driven radioresistance.

Therefore, I would suggest elaborating more on the following major points:

 Major:

·      Please include a more detailed section describing other relevant modes of tumor hypoxia within a solid tumor (e.g. acute, severe hypoxia, chronic cycling hypoxia) to the 2.2-Section explaining its documented effects on radioresistance, metabolism, acquired radioresistance. Furthermore, the authors should discuss the effects of Hypoxia in the view of proton vs. photon therapy.

·      Please mention/include and discuss recent work on the necessity for DNA repair pathways upon Photon and Proton irradiation

·      Please provide a more comprehensive view on the tumor cell metabolism contributing to radioresistance. Especially, the authors state, that tumors do have a glycolytic phenotype (line 273) contributing to radioresistance (line 286). However, recent work demonstrated that mitochondrial function exerts important effects on the radiosensitivity of cancer cells (e.g. Grasso et al, Front Pharmacol. 2020, PMID: 32231567 and Tang et al, J Exp Clin Cancer Res. 2018, PMID: 29688867, Krysztofiak et al., iScience 2021, PMID: 34825138

·      The authors mention the radiolysis of water a physical event taking place especially after irradiation with photons. However, the authors do not mention the fact, that photon irradiation triggers the formation of reactive oxygen specias (ROS) that damage cellular components, e.g., the DNA, and thereby contribute to the cytotoxic action of ionizing radiation (e.g. Dayal et al., J Cancer Res Ther, 2014, PMID: 25579513 and Panieri et al., Free Radic Biol Med, 2013, PMID: 23295411). Consequently, improved defense against cellular ROS by cellular antioxidant systems like glutathione will support resistance against the cytotoxic effects of ionizing radiation. Thus, cellular antioxidant capacity, e.g., cellular levels of GSH, will provide cells with the ability to protect themselves against intrinsic and extrinsic induction of ROS. Please discuss the important aspect of acquired radioresistance.

·      The authors discuss the variety of factors contributing to comparable phenotypes of radioresistance in line starting from 233. However the authors should discuss the concept of identification context-specific phenotypes for radiosensitization or radiationresistance (e.g. Daemen et al. 2020 CCR; doi: 10.1158/1078-0432.CCR-20-1835; Niemira et al. 2019 Cancers; doi: 10.3390/cancers12010037; Matschke et al., Biochem Soc Trans., 2021, PMID: 34110407)

·      Please correct or rephrase the sentence in line 275: “Consequently, the amount of molecular oxygen….”. It appears to me, that the amount of O2 drops due to the reduced function of the Oxphos in the tumor cells. However, recent studies even implied Oxphos inhibitors to overcome tumor hypoxia by increasing intracellular oxygen levels after inhibition of oxygen-consuming oxidative phosphorylation process with relevance for radiosensitization (e.g. Ashton TM et al., Clin Cancer Res 2018, PMID: 29420223).

Minor

·      Please correct the sentence in line 343 “The idea of improving…..with breathing oxygen. At 100%.” By removing the dot.

·      Please correct the sentence in line 357 by removing “72”: “The cellular…..DNA damage signaling pathways72, and …

·      Please correct the sentence in line 454 by removing the “75”: “…..Interest exist in the combination of…, including oxaliplatin75 as well….”

Author Response

Responses to Reviewer 2

The manuscript entitled “Biological mechanisms to reduce radioresistance and increase 2 the efficacy of radiotherapy: state of the art” discusses an important and relevant aspect of  basic concepts in radiobiology and molecular characteristics of radioresistant cancer cells. 

However, although being important, the authors only mention important effects of hypoxia and metabolism-driven radioresistance.

Therefore, I would suggest elaborating more on the following major points:

 Major:

1)Reviewer’s comment: Please include a more detailed section describing other relevant modes of tumor hypoxia within a solid tumor (e.g., acute, severe hypoxia, chronic cycling hypoxia) to the 2.2-Section explaining its documented effects on radioresistance, metabolism, acquired radioresistance. Furthermore, the authors should discuss the effects of Hypoxia in the view of proton vs. photon therapy.

Authors’ response: We welcome the reviewer’s suggestion by editing the section 2.2 on hypoxia. From line 272 onwards: ”The metabolism of cancer cells is another important factor that dictates the efficiency of curative protocols, including RT. Cancer cells rely on anaerobic and aerobic glycolysis rather than oxidative phosphorylation [Liberti and Localasale, 2016]  and for this reason, among others, can survive with respect to normal cells. Under glycolytic metabolism the amount of molecular oxygen (O2) within the tumour environment is reduced due to the disorganized nature of the vasculature [Begg and Tavassoli 2020] generating a condition of hypoxia, which comprehends three different categories within the solid tumor mass: chronic hypoxia, acute hypoxia, and cycling (or intermittent) hypoxia [Saxena and Jolly, 2019; Rakotomalala et al., 2021]. Chronic hypoxia is when cells experience insufficient O2 amount in a quasi-steady state, whereas acute and cycling hypoxia refer to temporal variations of pO2 [Bader et al., 2021]. Hypoxia is strictly associated with resistance of cancer cells to IR-induced cytotoxicity mainly because with low levels of oxygen the generation of ROS is reduced and consequently DNA damage is less [95; Begg and Tavassoli 2020]. Therefore, under hypoxic conditions cancer cells become less sensitive to radio-induced DNA damage and resist to radiotherapy, despite the activation of ATM signaling pathway [96]. Interestingly, such pathway resulted activated under hypoxic conditions even in non-irradiated cells, suggesting that ATM activation can occur in absence of DSBs [97]. Cancer cells can decrease endogenous ROS level by activating multiple antioxidant enzymes (i.e., glutathione reductase, superoxide dismutase, thioredoxin reductase, catalase) that provide antioxidant molecules to protect themselves against oxidative stress and can be responsible of their radioresistance [Liu et al., 2022; Flor et al., 2021; Jiang et al., 2018]. Under hypoxic conditions the hypoxia-inducible factor (HIF-1) is stabilized to activate a transcriptional program that stimulates glycolysis and down-regulates oxidative phosphorylation [100]. HIF-1 is constituted by the two subunits alpha and beta (HIF-1a and HIF-1b) whose dynamics varies within hypoxic cancer cells. HIF-1a stability increased during cyclic compared with chronic hypoxia, whereas HIF-1b is insensitive to variations in oxygen level, as recently reviewed in Saxena and Jolly [2019]. Once activated HIF-1 triggers a transcriptional pathway of hundreds of genes, including those of glucose metabolism, cell cycle regulation and proliferation, apoptosis, immune response, protection of tumor blood vessel and angiogenesis [Harada et al., 2016; Carmeliet et al., 1998; Jin et al., 2022; Wu et al., 2022], that ultimately allow the cellular adaptive response to hypoxia [102]. This consists of the formation of new blood vessels [103], the shift to anaerobic metabolism for cellular energy production, increased apoptosis and increased myeloid cells migration in inflamed areas” [Ziello et al., 2007].

Metabolic changes due to alterations in glucose and mitochondrial metabolic pathways contribute to radioresistance of cancer cells [McCann et al., 2021; Tang et al., 2018]. Alterations in the expression level of key proteins and molecules of glycolytic process (i.e., glucose transporters, lactic acid, pyruvate kinase, hexokinases) and mitochondrial function (i.e., adenosine monophosphate family protein 3A, SIRT3, mitochondrial MAPK phosphatase) have been reported in radioresistant cancer cells [Tang et al., 2018]. A transient decrease in mitochondrial function has been observed in cancer cell lines shortly after exposure to IR which correlates with oxidizing effects of IR [Krysztofiak et al., 2021] and could be considered for tumor radioresistance. Indeed, colon and lung cancer cells that shifted their metabolism to glycolysis because of mitochondria depletion manifested radioresistance [Shi et al., 2021; Wei et al., 2018]. On the other hand, an increase in mitochondrial abundance and a high oxidative metabolism have been reported in radioresistant vs. radiosensitive human HNSCC cells [Grasso et al., 2020]. To complicate the picture of the heterogeneity of tumor cells, some cancer subtypes exhibit high oxidative phosphorylation which can be considered a target for clinical application [Ashton et al., 2018]. To add further complexity, hypoxia can impact on IR-induced cytotoxicity in a radiation quality-related manner. Carbon ions (high-LET radiation) resulted efficient in counteracting radioresistance of HNSCC cells since they exert their action regardless of oxygen, in contrast to photons (low LET radiation) whose action is dependent on oxygen concentration [Wozny et al., 2020]. Interestingly, the acquisition of radioresistance in vitro in cancer cells can be exerted by repeated irradiation with photons (X-rays) but not with particle (c-ions) [Sato et al., 2019].

2)Reviewer’s comment: Please mention/include and discuss recent work on the necessity for DNA repair pathways upon Photon and Proton irradiation

Authors’ response: We thank the reviewer for this suggestion. Line 99 of the revised manuscript, we add as follows: ”IR-induced DSBs are the lesions that more contribute to the death of cancer cells but can be efficiently repaired either by HR or NHEJ. Deficiencies in essential proteins of NHEJ and HR pathway affect radiosensitivity of both normal and cancer cells, highlighting the central role of NHEJ in DSB repair [Bee et al., 2013; Piotto et al., 2018; Szymonowicz et al., 2020]. Notably, the two DSB repair systems operate differently in cancer cells in response to proton- and photon-irradiation, highlighting the preference of proton-induced DNA damage toward HR in human cancer cells [Fontana et al., 2015; Gerelchuluun et al., 2015; Szymonowicz et al., 2020]. The reason is related to the different nature of DNA damage, which is more complex and clustered after proton-irradiation [Ray et al., 2018] and could impede the activity of Ku protein of NHEJ pathway [Watanabe et al., 2011]. However, when both systems are active (i.e., in late S and G2 phases) NHEJ can operate when the lesion is not too complex, otherwise there is a switch toward HR [Szymonowicz et al., 2020]. 

3)Reviewer’s comment: Please provide a more comprehensive view on the tumor cell metabolism contributing to radioresistance. Especially, the authors state, that tumors do have a glycolytic phenotype (line 273) contributing to radioresistance (line 286). However, recent work demonstrated that mitochondrial function exerts important effects on the radiosensitivity of cancer cells (e.g. Grasso et al, Front Pharmacol. 2020, PMID: 32231567 and Tang et al, J Exp Clin Cancer Res. 2018, PMID: 29688867, Krysztofiak et al., iScience 2021, PMID: 34825138.

Authors’ response: Following the reviewer’s suggestion we edited section 2.2. as follows:” Metabolic changes due to alterations in glucose and mitochondrial metabolic pathways contribute to radioresistance of cancer cells [McCann et al., 2021; Tang et al., 2018]. Alterations in the expression level of key proteins and molecules of glycolytic process (i.e., glucose transporters, lactic acid, pyruvate kinase, hexokinases) and mitochondrial function (i.e., adenosine monophosphate family protein 3A, mitochondrial MAPK phosphatase) have been reported in radioresistant cancer cells [Tang et al., 2018]. A transient decrease in mitochondrial function has been observed in cancer cell lines shortly after exposure to IR which correlates with oxidizing effects of IR [Krysztofiak et al., 2021] and could be considered for tumor radioresistance. Indeed, colon and lung cancer cells that shifted their metabolism to glycolysis because of mitochondria depletion manifested radioresistance [Shi et al., 2021; Wei et al., 2018]. On the other hand, an increase in mitochondrial abundance and a high oxidative metabolism have been associated with the acquired radioresistance of human laryngeal squamous cell carcinoma cells [Grasso et al., 2020]”.

4)Reviewer’s comment: The authors mention the radiolysis of water a physical event taking place especially after irradiation with photons. However, the authors do not mention the fact, that photon irradiation triggers the formation of reactive oxygen specias (ROS) that damage cellular components, e.g., the DNA, and thereby contribute to the cytotoxic action of ionizing radiation (e.g. Dayal et al., J Cancer Res Ther, 2014, PMID: 25579513 and Panieri et al., Free Radic Biol Med, 2013, PMID: 23295411). Consequently, improved defense against cellular ROS by cellular antioxidant systems like glutathione will support resistance against the cytotoxic effects of ionizing radiation. Thus, cellular antioxidant capacity, e.g., cellular levels of GSH, will provide cells with the ability to protect themselves against intrinsic and extrinsic induction of ROS. Please discuss the important aspect of acquired radioresistance.

Authors’ response: We thank the reviewer for this comment. In the Introduction section, line 44, we edited as follows:” In addition to direct DNA damage, originated by the physical interaction between IR and DNA helix, water radiolysis occurs in irradiated cells and the reactive oxygen species (ROS) produced give rise to oxidative stress and DNA damage generation [Beheshti et al., 2021; Dayal et al., 2014] that contribute to cytotoxic effects of IR [Panieri et al., 2013].

Regarding the “antioxidant capacity of cancer cells”, we commented on this point in section 2.2., line 300 of revised manuscript: “Cancer cells can decrease endogenous ROS level by activating multiple antioxidant enzymes (i.e., glutathione reductase, superoxide dismutase, thioredoxin reductase, cat-alase) that provide antioxidant molecules to protect themselves against oxidative stress and can be responsible of the acquired radioresistance [Panieri et al., 2013; Liu et al., 2022].”

About “acquired radioresistance” in section 2.2 line 78, we described the main differences between intrinsic and acquired radioresistance. We edited as follows:” Radioresistance can be intrinsic or acquired, the former being mainly associated with molecular features of cancer cells, such as dysregulation in DDR pathway, chronically activated proliferative, invasive, antiapoptotic signaling pathway, and IR-induced ROS level [10-11 Dayal et al., 2014; Panieri et al., 2013], the second being mainly associated with modifications in the cell compartment within the tumor environment such as enrichments in cancer stem cells (CSCs), epithelial to mesenchymal transition, and activation of transcription factors involved in CSC stemness maintenance [12; Galeaz et al., 2021]. Acquired radioresistance established following repeated IR fractions was associated also with enhancement in DNA repair capacity and autophagic cell death [Kuwahara et al., 2009; 2017].”

About acquired radioresistance, line 58 of submitted first version we stated that “In general, it is thought that DNA methylation changes occur during the phenomenon of acquired radioresistance”.

5) Reviewer’s comment: The authors discuss the variety of factors contributing to comparable phenotypes of radioresistance in line starting from 233. However the authors should discuss the concept of identification context-specific phenotypes for radiosensitization or radiationresistance (e.g. Daemen et al. 2020 CCR; doi: 10.1158/1078-0432.CCR-20-1835; Niemira et al. 2019 Cancers; doi: 10.3390/cancers12010037; Matschke et al., Biochem Soc Trans., 2021, PMID: 34110407)

Authors’ response: In section 2.1, line 232 of the revised manuscript we referred to genomic studies reporting changes associated with tumor radioresistance.  The suggested references (Daemen et al.,2021 and Niemira et al.,2019) demonstrate the ability transcription signatures to classify different tumor subtypes in view of targeted therapy although do not refer to tumor radioresistance. However, to meet the reviewer’s suggestion we added the following phrase, line 235:” Recent studies reported that transcriptional profiles can be useful to classify subtypes in early-advanced stage lung adenocarcinoma and to identify the better therapeutic strategies leveraging in vitro, in vivo, and clinical trial data [Daemen et al., 2021]. Also, transcriptome profiles can be associated with histological subtypes of NSCLC [Niemira et al., 2019].  However, the genetic and epigenetic scenario in radioresistant tumors is complicated to describe since, if from one side genetic and epigenetic signatures can drive tumorigenesis, on the other hand radiotherapy itself can trigger long-term genetic and epigenetic changes in cancer cells that in turn affect radioresistance. Few studies have analyzed the role of ncRNAs in radioresistance and metabolic phenotypes of radioresistant cancer cells [Wu et al.2022; Matschke et al., 2021]. While the differentially expressed genes and ncRNAs following IR have been identified in numerous tumors, what is still little known is the relationship between specific genes/ncRNAs and radioresistance. The difficulty consists in the wide range of interconnected biological pathways that are either at the basis of radio-resistance and are activated in response to radiation.  Additional studies employing genome-wide approach together with clinical treatments will be required for future investigations on tumor radioresistance.

6) Reviewer’s comment: Please correct or rephrase the sentence in line 275: “Consequently, the amount of molecular oxygen….”. It appears to me, that the amount of O2 drops due to the reduced function of the Oxphos in the tumor cells. However, recent studies even implied Oxphos inhibitors to overcome tumor hypoxia by increasing intracellular oxygen levels after inhibition of oxygen-consuming oxidative phosphorylation process with relevance for radiosensitization (e.g. Ashton TM et al., Clin Cancer Res 2018, PMID: 29420223).

Authors’ response: We thank the reviewer for this criticism. We edited the whole part as follows:” The metabolism of cancer cells is another important factor that dictates the efficiency of curative protocols, including RT. Cancer cells rely on anaerobic and aerobic glycolysis rather than oxidative phosphorylation [Liberti and Localasale, 2016] and for this reason, among others, can survive with respect to normal cells. Under glycolytic metabolism the amount of molecular oxygen (O2) within the tumour environment is reduced due to the disorganized nature of the vasculature [Begg and Tavassoli 2020] generating a condition of hypoxia, which comprehends three different categories within the solid tumor mass: chronic hypoxia, acute hypoxia, and cycling (or intermittent) hypoxia [Saxena and Jolly, 2019; Rakotomalala et al., 2021]. Chronic hypoxia is when cells experience insufficient O2 amount in a quasi-steady state, whereas acute and cycling hypoxia refer to temporal variations of pO2 [Bader et al., 2020]. Hypoxia is strictly associated with resistance of cancer cells to IR-induced cytotoxicity mainly because with low levels of oxygen the generation of ROS is reduced and consequently DNA damage is less [95; Begg and Tavassoli 2020]. Therefore, under hypoxic conditions cancer cells become less sensitive to radio-induced DNA damage and resist to radiotherapy, despite the activation of ATM signaling pathway [96]. Interestingly, such pathway resulted activated under hypoxic conditions even in non-irradiated cells, suggesting that ATM activation can occur in absence of DSBs [97]. Cancer cells can decrease endogenous ROS level by activating multiple antioxidant enzymes (i.e., glutathione reductase, superoxide dismutase, thioredoxin reductase, catalase) that provide antioxidant molecules to protect themselves against oxidative stress and can be responsible of the acquired radioresistance [Panieri et al., 2013; Liu et al., 2022]. Under hypoxic conditions the hypoxia-inducible factor (HIF-1) is stabilized to activate a transcriptional program that stimulates glycolysis and down-regulates oxidative phosphorylation [100]. HIF-1 is constituted by the two subunits alpha and beta (HIF-1a and HIF-1b) whose dynamics varies within hypoxic cancer cells. HIF-1a stability increased during cyclic compared with chronic hypoxia, whereas HIF-1b is insensitive to variations in oxygen level, as recently reviewed in Saxena and Jolly [2019]. Once activated HIF-1 triggers a transcriptional pathway of hundreds of genes, including those of glucose metabolism, cell cycle regulation and proliferation, apoptosis, immune response, protection of tumor blood vessel and angiogenesis [Al Taeemeni et al., 2019; Harada et al., 2016; Carmeliet et al., 1998; Jin et al., 2022]. Metabolic changes due to alterations in glucose and mitochondrial metabolic pathways contribute to radioresistance of cancer cells [McCann et al., 2021; Tang et al., 2018]. Alterations in the expression level of key proteins and molecules of glycolytic process (i.e., glucose transporters, lactic acid, pyruvate kinase, hexokinases) and mitochondrial function (i.e., adenosine monophosphate family protein 3A, SIRT3, mitochondrial MAPK phosphatase) have been reported in radioresistant cancer cells [Tang et al., 2018]. A transient decrease in mitochondrial function has been observed in cancer cell lines shortly after exposure to IR which correlates with oxidizing effects of IR [Krysztofiak et al., 2021] and could be considered for tumor radioresistance. Indeed, colon and lung cancer cells that shifted their metabolism to glycolysis because of mitochondria depletion manifested radioresistance [Shi et al., 2021; Wei et al., 2018]. On the other hand, an increase in mitochondrial abundance and a high oxidative metabolism have been reported in radioresistant vs. radiosensitive human HNSCC cells [Grasso et al., 2020]. To complicate the picture of the heterogeneity of tumor cells, some cancer subtypes exhibit high oxidative phosphorylation which can be considered a target for clinical application [Ashton et al., 2018]. To add further complexity, hypoxia can impact on IR-induced cytotoxicity in a radiation quality-related manner. Carbon ions (high-LET radiation) resulted efficient in counteracting radioresistance of HNSCC cells since they exert their action regardless of oxygen, in contrast to photons (low LET radiation) whose action is dependent on oxygen concentration [Wozny et al., 2020]. Interestingly, the acquisition of radioresistance in vitro in cancer cells can be exerted by repeated irradiation with photons (X-rays) but not with particle (c-ions) [Sato et al., 2019].

Minor

  • Please correct the sentence in line 343 “The idea of improving…..with breathing oxygen. At 100%.” By removing the dot.

Authors’ response: Done

  • Please correct the sentence in line 357 by removing “72”: “The cellular…..DNA damage signaling pathways72, and …

Authors’ response: Done

  • Please correct the sentence in line 454 by removing the “75”: “…..Interest exist in the combination of…, including oxaliplatin75 as well….”

Authors’ response: Done

Round 2

Reviewer 1 Report

Busato et al. present the revised version of their review entitled ‘Biological mechanisms to reduce radioresistance and increase the efficacy of radiotherapy: state of the art’.

Some of the comments were well-addressed others were not.

However, many aspects of the manuscript need extensive improvement; I cannot support a final publication.

In general, the text is not well structured, as it should be for a review paper.

A clear strategy for the text structure and sections would improve the presentation of the information and make it easier to read. My main concern is still the fact that the authors want to give an overview of the state of the art concerning radiosensitization. However, many relevant and promising new approaches are completely ignored. This was already commented on in the first version without receiving a proper response from the authors (Comment #10). Often, the authors simply summarize data and sources and do not critically evaluate them to provide a clear picture of the state of knowledge on this topic and future perspectives.

The references are messed up and have, at least partly, the numbers of the first version. Partially unnumbered (Line 217). So no proper review is possible. In the first version, the references ended at no. 125, in the body of the text they went up to no. 144.

There are several incomplete or grammatically incorrect sentences. The text should be readable and of high quality, so an improvement of the English language is necessary.

Some examples:

·         Line 54: ‘Recent data show that cancer senescent cells trigger…’ should read ‘Recent data show that senescent cancer cells trigger…’

·         Line 111: ‘…critical role in DNA damage signaling and repairing..’, should read ‘…critical role in DNA damage signaling and repair..’

·         Line 334: ‘Interestingly, such pathway resulted activated under hypoxic conditions…’

Some major comments:

1Line 37: Not only in G1 cells. Also in proliferating cells, 80% of DSBs are repaired by NHEJ, only about 20% by HR in the S phase and the late repair component of the G2 phase.

2Line 39: In discussing the various DNA repair pathways, an overview of the different types of IR-induced DNA damage and their extent and lethality should be provided beforehand. Why did the authors delete the first sentence dealing with IR-induced DNA damage and instead start with the cellular response to it?

3Line 63: This section does not deal, first at all, with ‘Molecular and cellular features associated with radioresistance’ but with the basics of RT. There should be a section before that properly summarizes the basics of radiobiology and radiotherapy.

4Line 82: The explanation of the RBE is incomplete.

'Comparing two radiation qualities'; so comparing a test radiation (e.g. protons or c-ions) to photons (X-rays or gamma-rays) as the standard reference radiation.

5Line 400: A brief, appropriate introduction to the topic should be given here.

                Why are inhibitors of DNA repair mentioned in this paragraph?

A separate section should be dedicated to inhibitors of DNA repair and cell cycle regulation as potent radiosensitizers and corresponding new therapeutic approaches and (pre-)clinical studies should be discussed.

6Line 476: The authors should at least give a brief introduction to the role of cytostatic drugs in the radiosensitization of tumors. These are listed here rather abruptly since the role of chemotherapy or radiochemotherapy was not mentioned earlier in the manuscript at all.

E.g., the different classes (alkylating agents, antimetabolites, mitotic spindle inhibitors, topoisomerase inhibitors, and others) and their main mechanisms of action.

For the cytostatic drugs, the general mechanism of action should be mentioned first, and then how it interacts with the effects of IR.

7 References must be provided for quite some statements:

Lines 499, 470, 474, 535 (for non-mammalian),

8 Line 594:  ‘includes checkpoint inhibitors’, should read ‘includes immune checkpoint inhibitors’

9Line 597: The authors should mention the IR-associated release of DAMPs representing a very important immunostimulatory factor for immuno-oncology.

1 Line 331 and the following: The authors get entangled in contradictions.

Which effect is responsible for hypoxia-mediated radioresistance and the OER? A reduced generation of ROS and thus DNA damage by indirect ionization or the oxygen fixation hypothesis? The effect is justified sometimes in this way, sometimes in that way.

1Line 334: What about the bystander effect previously mentioned by the authors? Bystander cells also show elevated levels of DNA damage measured as gH2AX foci, micronuclei, sister chromatid exchanges, etc.

1Previous comment on line 84: IR has been introduced in the abstract and has been used already in line 28. Check the manuscript thoroughly for the introduction and use of acronyms, e.g., DSB in line 320, RBE in line 510….

Authors’ response: we corrected.

Not properly addressed. Still, acronyms are introduced but not used consistently applied, e.g. RT: lines 107, 168, 180…., IR: lines 187, 538…..

Author Response

Busato et al. present the revised version of their review entitled ‘Biological mechanisms to reduce radioresistance and increase the efficacy of radiotherapy: state of the art’.

Some of the comments were well-addressed others were not. However, many aspects of the manuscript need extensive improvement; I cannot support a final publication.

In general, the text is not well structured, as it should be for a review paper.

-A clear strategy for the text structure and sections would improve the presentation of the information and make it easier to read. My main concern is still the fact that the authors want to give an overview of the state of the art concerning radiosensitization. However, many relevant and promising new approaches are completely ignored. This was already commented on in the first version without receiving a proper response from the authors (Comment #10). Often, the authors simply summarize data and sources and do not critically evaluate them to provide a clear picture of the state of knowledge on this topic and future perspectives.

Authors’ response. At the end of the Introduction, we added a sentence to help the reader understand the logical structure of the manuscript which is composed of a first section on the basic concepts of radiobiology and the molecular/cellular characteristics of radioresistant cancer cells (paragraph 2) and a second section on present and future targets against radioresistant tumors both in pre-clinical and clinical setting (paragraph 3).

Regarding the relevant new approaches cited by the reviewer, we are aware of the abundant new strategies and tried to select the most studied ones and, most of all, those which are closer to clinicians. This selection comes from a personal point of view and, conscious of our limitations, we have added a new section (3.2.7) on DNA-repair and cell cycle inhibitors.

-The references are messed up and have, at least partly, the numbers of the first version. Partially unnumbered (Line 217). So no proper review is possible. In the first version, the references ended at no. 125, in the body of the text they went up to no. 144.

Authors’ response. The reviewer is right, in the first version of the manuscript references ended at number 125 because in the submission process the uploaded version was not exactly the very final one. However, we had noticed the lack and we immediately refresh the submitted paper but probably the system did not save the changes. We apologize for this. The revised manuscript contains the updated references.

-There are several incomplete or grammatically incorrect sentences. The text should be readable and of high quality, so an improvement of the English language is necessary.

Some examples:

  • Line 54: ‘Recent data show that cancer senescent cells trigger…’ should read ‘Recent data show that senescent cancer cells trigger…’
  • Line 111: ‘…critical role in DNA damage signaling and repairing..’, should read ‘…critical role in DNA damage signaling and repair..’
  • Line 334: ‘Interestingly, such pathway resulted activated under hypoxic conditions…

 Authors’ response. We corrected, thank you.

Some major comments:

1Line 37: Not only in G1 cells. Also in proliferating cells, 80% of DSBs are repaired by NHEJ, only about 20% by HR in the S phase and the late repair component of the G2 phase.

Authors’ response. The reviewer is right. What we want saying is that quiescent cells can rely only on NHEJ since they do not have sister chromatids for carrying on HR. However, we understand that the sentence as it is can be misunderstood, therefore we changed as follows: “In mammalian cells, both quiescent and proliferating, non-homologous end joining (NHEJ) is the predominant repair pathway for DSBs, in contrast to homologous recombination (HR) that intervenes only when sister chromatids are available in S-G2 phases of cell cycle [Daley et al., 2015].

2Line 39: In discussing the various DNA repair pathways, an overview of the different types of IR-induced DNA damage and their extent and lethality should be provided beforehand. Why did the authors delete the first sentence dealing with IR-induced DNA damage and instead start with the cellular response to it?

Authors’ response. We thank the reviewer for this suggestion. We added an overview of the different types of IR-induced DNA damage. Line 34, “Different types of DNA damage can originate from direct and indirect effect of IR. Base damage, single and double strand breaks (SSBs and DSBs) originate when IR hits the DNA molecule, that have been quantified respectively in 850 pyrimidine lesions, 450 purine lesions, 1000 SSBs and 20-40 DSBs/cell/Gy with g-radiation in mammalian cells [Lomax et al., 2013]. In addition to direct DNA damage, originated by the physical interaction between IR and DNA helix, water radiolysis occurs in irradiated cells and the reactive oxygen species (ROS) produced give rise to oxidative stress and DNA damage generation [4; Dayal et al., 2014] that contribute to cytotoxic effects of IR [Panieri et al., 2013]. Oxidative DNA damage consists in oxidized bases such as 8-Oxo-7,8-dihydro-2′-deoxyguanosine (8-oxodGuo), oxidized pyrimidine derivatives (i.e.  thymine glycol and 5,6 dihydrouracil), oxidized base-derived apurinic/apyrimidinic sites and SSBs [Beheshti et al., 2021]. According to the nature of IR-induced DNA damage different pathways come into play to repair the lesion and restore cell functionality. In mammalian cells, both quiescent and proliferating, non-homologous end joining (NHEJ) is the predominant repair pathway for DSBs, in contrast to homologous recombination (HR) that intervenes only when sister chromatids are available in S-G2 phases of cell cycle [Daley et al., 2015]. Base excision repair (BER) and nucleotide excision repair (NER) are also active in repairing radio-induced DNA damage in the forms of SSBs and base/nucleotide lesions [3]. Ultimately, irradiated cells can affect the physiology of non-directly irradiated cells by the “bystander effect” [5]. DDR pathway functionality is intimately associated with radioresistance of human cancer cells and new approaches have been explored to counteract this phenomenon through the targeting of different pathways.

-The choice to move the sentence from the introduction to paragraph 2 line 106, where it stands as:” Radiotherapy causes the death of cancer cells mainly by IR-induced DSBs which are the most important cytotoxic lesions induced by IR” is because here it is more linked to the part on alterations on DSB repair protein and radioresistance. However, in the introduction we left the very first phrase: “Radiotherapy (RT) causes the death of cancer cells mainly by ionizing radiation (IR)-induced DNA damage”. We hope that the reviewer agrees.

3Line 63: This section does not deal, first at all, with ‘Molecular and cellular features associated with radioresistance’ but with the basics of RT. There should be a section before that properly summarizes the basics of radiobiology and radiotherapy.

Authors’ response. We welcome the reviewer’s suggestion by adding a sub-section entitled: “Basics of radiobiology and radiotherapy”.

4Line 82: The explanation of the RBE is incomplete.

'Comparing two radiation qualities'; so comparing a test radiation (e.g. protons or c-ions) to photons (X-rays or gamma-rays) as the standard reference radiation.

Authors’ response. We thank the reviewer for this important detail. We corrected as follows:” RBE is defined as the ratio of doses to reach the same level of effect when comparing a test radiation (i.e., protons or c-ions) and photons (X-rays or gamma rays) as the standard reference radiation [13, 15].”

5Line 400: A brief, appropriate introduction to the topic should be given here.

                Why are inhibitors of DNA repair mentioned in this paragraph?

A separate section should be dedicated to inhibitors of DNA repair and cell cycle regulation as potent radiosensitizers and corresponding new therapeutic approaches and (pre-)clinical studies should be discussed.

Authors’ response. Thank you, as said above, we have added a specific section (3.2.7.) entitled ”DNA-repair and cell cycle inhibitors”.

6Line 476: The authors should at least give a brief introduction to the role of cytostatic drugs in the radiosensitization of tumors. These are listed here rather abruptly since the role of chemotherapy or radiochemotherapy was not mentioned earlier in the manuscript at all.

E.g., the different classes (alkylating agents, antimetabolites, mitotic spindle inhibitors, topoisomerase inhibitors, and others) and their main mechanisms of action.

For the cytostatic drugs, the general mechanism of action should be mentioned first, and then how it interacts with the effects of IR.

Authors’ response. We have added the mechanism of action, where not specified.

7 References must be provided for quite some statements:

Lines 499, 470, 474, 535 (for non-mammalian),

Authors’ response. We have added the following references:

Line 499:

“Wang Y, Deng W, Li N, Neri S, Sharma A, Jiang W, Lin SH. Combining Immunotherapy and Radiotherapy for Cancer Treatment: Current Challenges and Future Directions. Front Pharmacol. 2018 Mar 5;9:185. doi: 10.3389/fphar.2018.00185. PMID: 29556198; PMCID: PMC5844965.”

Line 470:

“Wedge SR, Porteous JK, Glaser MG, Marcus K, Newlands ES. In vitro evaluation of temozolomide combined with X-irradiation. Anticancer Drugs. 1997 Jan;8(1):92-7. doi: 10.1097/00001813-199701000-00013. PMID: 9147618.”

“Whoon Jong Kil, David Cerna, William E. Burgan, Katie Beam, Donna Carter, Patricia S. Steeg, Philip J. Tofilon, Kevin Camphausen; In vitro and In vivo Radiosensitization Induced by the DNA Methylating Agent Temozolomide. Clin Cancer Res 1 February 2008; 14 (3): 931–938. https://doi.org/10.1158/1078-0432.CCR-07-1856.”

Line 474:

Ibrahim Khan, Khalid Saeed, Idrees Khan; Nanoparticles: Properties, applications and toxicities. Arabian Journal of Chemistry, Volume 12, Issue 7, 2019 ; Pages 908-931, https://doi.org/10.1016/j.arabjc.2017.05.011.

Line 535: The reference De Meerleer et al.,2014 was already present.

8 Line 594:  ‘includes checkpoint inhibitors’, should read ‘includes immune checkpoint inhibitors’

Authors’ response. We edited, thank you.

9Line 597: The authors should mention the IR-associated release of DAMPs representing a very important immunostimulatory factor for immuno-oncology.

Authors’ response. We thank the reviewer for pointing out this comment. In section 3.4 immunomodulator, line 597 we edited as follows: “….iiii) by releasing damage-associated molecular patterns (DAMPs) that can activate immune system against tumor cells (Ashrafizadeh M, Farhood B, Eleojo Musa A, Taeb S, Najafi M. Damage-associated molecular patterns in tumor radiotherapy. Int Immunopharmacol. 2020 Sep;86:106761. doi: 10.1016/j.intimp.2020.106761. Epub 2020 Jul 3. PMID: 32629409.)”.

1 Line 331 and the following: The authors get entangled in contradictions.

Which effect is responsible for hypoxia-mediated radioresistance and the OER? A reduced generation of ROS and thus DNA damage by indirect ionization or the oxygen fixation hypothesis? The effect is justified sometimes in this way, sometimes in that way.

Authors’ response. The effect responsible for hypoxia-mediated radioresistance is the low amount of oxygen and consequently the less oxidative DNA damage caused by ROS. As stated by Grimes and Partridge, 2015 “In essence, damage produced by free-radicals can be restored under hypoxia but is ‘fixed’ (made permanent and irreparable) when molecular oxygen is present.”

We welcome the reviewer’s comment, and we made clearer this section by editing as follows:” Hypoxia is strictly associated with resistance of cancer cells to IR-induced cytotoxicity mainly because with low levels of oxygen the generation of ROS is reduced and consequently DNA damage is less [95; Begg and Tavassoli 2020]. Conversely, well-oxygenated cells respond better to RT by a factor of 2.5-3, as a consequence of the oxygen enhancement ratio, which is an important component in photon and low LET therapy [Grimes and Partridge, 2015]. The oxygen effect is most commonly explained by the oxygen fixation hypothesis, which postulates that radical-induced DNA damage can be ‘fixed’ by molecular oxygen, rendering DNA damage difficult or impossible to for the cell to repair”.

We hope to have properly addressed your question.

1Line 334: What about the bystander effect previously mentioned by the authors? Bystander cells also show elevated levels of DNA damage measured as gH2AX foci, micronuclei, sister chromatid exchanges, etc.

Authors’ response. We mentioned “bystander effect” only in Introduction section, as a modality of irradiated cells to affect the physiology of non-directly irradiated cells. Though we agree with what has been written about the effect on bystander cells, we have some difficulty in answering as line 334 refers to other aspects and we’re not able to precisely understand the involved sentence. In general, as the article is centered on tumors, we simply cited the bystander effect. However, we added the reviewer’s suggestion in that point. “Ultimately, irradiated cells can affect the physiology of non-directly irradiated cells by the “bystander effect” which is a cause of elevated levels of DNA damage measured as g-H2AX foci, micronuclei, sister chromatid exchanges [5].”

1Previous comment on line 84: IR has been introduced in the abstract and has been used already in line 28. Check the manuscript thoroughly for the introduction and use of acronyms, e.g., DSB in line 320, RBE in line 510….

Authors’ response: we corrected.

Not properly addressed. Still, acronyms are introduced but not used consistently applied, e.g. RT: lines 107, 168, 180…., IR: lines 187, 538…..

Authors’ response. We carefully checked the whole manuscript. We hope to have corrected all acronyms. However, sometimes we would like to use the entire word radiotherapy rather than its acronym (i.e., lines 72, 697, 705, 737, 761, 798).

Reviewer 2 Report

The revised version of the manuscript substantially improved and is suitable for publication. Thanks to the authors for extensively addressing the comments.

Author Response

Thank you

Round 3

Reviewer 1 Report

All of my concerns were properly addressed by the authors and I recommend the publication of the review in its present form.